# Make Still Further Progress:
# Chain of Thoughts for Tabular Data Leaderboard

## Abstract

Tabular data, a fundamental data format in machine learning, is predominantly utilized in competitions and real-world applications. The performance of tabular models—such as gradient boosted decision trees and neural networks—can vary significantly across datasets due to differences in feature distributions and task characteristics. Achieving top performance on each dataset often requires specialized expert knowledge. To address this variability, practitioners often aggregate the predictions of multiple models. However, conventional aggregation strategies typically rely on static combination rules and lack instance-level adaptability. In this work, we propose an in-context ensemble framework for tabular prediction that leverages large language models (LLMs) to perform dynamic, instance-specific integration of external model predictions. Without access to raw tabular features or semantic information, our method constructs a context around each test instance using its nearest neighbors and the predictions from a pool of external models. Within this enriched context, we introduce Chain of Tabular Thoughts ($CoT^2$), a prompting strategy that guides LLMs through multi-step, interpretable reasoning, making still further progress toward expert-level decision-making. Experimental results show that our method outperforms well-tuned baselines and standard ensemble techniques across a wide range of tabular datasets.

## 1 Introduction

Tabular data holds a pivotal position in the field of machine learning, primarily because of its organized and accessible format. In many competitions and real-world applications, ranging from financial forecasting Addo et al. (2018) to healthcare diagnostics Hassan et al. (2020), tabular data serves as the primary data format. Recently, Gradient Boosted Decision Trees (GBDTs) Chen & Guestrin (2016); Ke et al. (2017); Prokhorenkova et al. (2018) and Neural Networks (NN) Gorishniy et al. (2021); Ye et al. (2023); Borisov et al. (2022) are two of the most commonly explored methods for tabular data learning Jiang et al. (2025a). However, although GBDTs often outperform NNs across many datasets Grinsztajn et al. (2022), the diverse nature of tabular data tasks implies that either method could be the most or least effective choice for a specific dataset McElfresh et al. (2023); Ye et al. (2024). In practice, achieving high accuracy often requires expert-level tuning and the integration of multiple models. For instance, top solutions in machine learning competitions frequently adopt ensemble strategies designed by experienced practitioners.

Large Language Models (LLMs)Achiam et al. (2023); Brown et al. (2020) have achieved remarkable success across a range of domains, including question answeringYu et al. (2023), code generation Zheng et al. (2024), and scientific reasoning Prabhakar et al. (2025). However, LLMs' application to tabular data prediction remains limited. This is primarily because tabular tasks often involve reasoning over numerical values Manikandan et al. (2023)—such as predicting house prices based on features like square footage and number of bedrooms—where LLMs typically underperform. Current research on applying LLMs to tabular data remains limited and is mostly constrained to datasets with comprehensive textual descriptions. Existing studies can be broadly categorized into two main approaches: One line of work directly converts tabular instances into text prompts using feature descriptions, allowing the LLM to act as a predictor Dinh et al. (2022). This approach is heavily dependent on the availability and quality of textual descriptions and often struggles with numerical precision. The other line of research uses LLMs to support traditional tabular

pipelines by automating steps like data cleaning Bendinelli et al. (2025), feature engineering Hollmann et al. (2023b), and hyperparameter tuning Zhang et al. (2024). However, the effectiveness of these methods remains fundamentally constrained by the richness and accessibility of semantic information. In many practical scenarios—especially those involving sensitive data or proprietary systems—such semantic information, including feature names or task-level descriptions, may be unavailable or inaccessible. This limitation hinders the deployment of LLM-based methods that rely on explicit textual representations. Motivated by this, we pose the following question:

> *When there are no textual descriptions, can we transform the LLM into a competition expert,*
> *leveraging its robust reasoning abilities to make predictions with minimal computational cost?*

To address the question above, our central idea is to empower LLMs to act like human experts in machine learning competitions: rather than directly accessing raw features, the LLM integrates multiple model predictions at the instance level, forming a dynamic ensemble guided by contextual knowledge. This approach leverages the LLMs' general reasoning capabilities to selectively synthesize and reconcile external predictions, much like how practitioners deliberate over conflicting model outputs when making final decisions. To realize this goal, we identify three key challenges:

First, *where does the knowledge come from*? In many real-world scenarios, the semantic richness of tabular data is often limited. For instance, features may consist solely of numerical outputs from multiple sensors, or data privacy concerns may restrict access to descriptive information. Thus, a critical objective is to construct a context that conveys essential predictive signals while avoiding reliance on raw features or textual descriptions. Unlike domains where data is inherently sequential and context-rich, tabular data

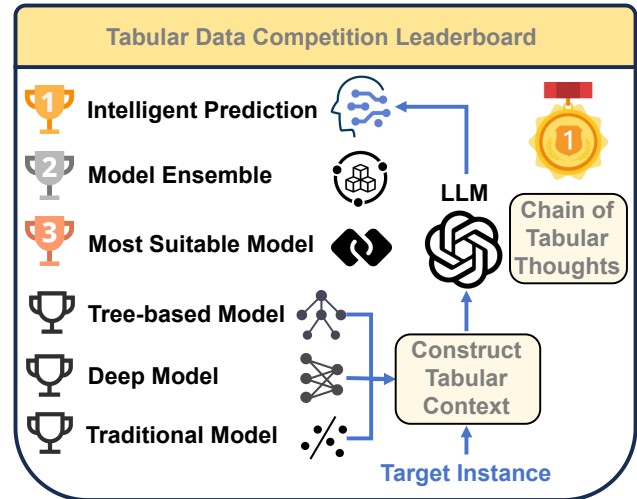

Figure 1: CoT$^2$ utilizes the expert knowledge of LLMs to create an intelligent ensemble of tabular models, making still further progress.

lacks natural contextual structure. To overcome this, we construct a tabular context for each target instance. We first identify local neighbors of the target and gather predictions from multiple external models within this neighborhood, combined with other non-semantic dataset information. This synthesized context is then used as a prompt to the LLM, which generates a final prediction. However, our empirical analysis shows that LLMs do not inherently interpret such context effectively without further guidance.

Second, *how do we guide the LLM to think*? Simply exposing the LLM to tabular contexts is not enough—we need to guide it to reason like an expert. Inspired by the Chain of Thought (CoT) Wei et al. (2022), we introduce a structured reasoning process tailored for tabular data: the Chain of Tabular Thoughts (CoT$^2$). CoT$^2$ decomposes the prediction process into multiple analytical steps, such as identifying outliers and selecting appropriate models, leveraging the interactions among neighboring instances and their associated predictions from external models. By guiding the LLM through this step-by-step reasoning, we enable it to detect anomalies and select the best models for the local neighborhood. CoT$^2$ compensates for the LLMs' limited sensitivity to raw numerical values, helping it make effective and interpretable predictions like a machine learning competition expert Figure 1.

After equipping LLMs with carefully constructed tabular contexts and chains of thought, the third challenge is *minimizing inference cost*. Since LLM-based methods require running inference for each target instance Dinh et al. (2022); Hegselmann et al. (2023); Gardner et al. (2024), reducing the number of instances that invoke LLMs is crucial for practical deployment. In real-world settings, many instances are relatively simple—multiple external models already yield consistent and accurate predictions for them. We identify such cases by measuring the agreement among external model outputs and bypass LLM processing when sufficient consensus

is observed. As a result, LLMs are reserved for more challenging instances, where model disagreement indicates greater uncertainty. This selective strategy significantly reduces computational overhead while preserving the benefits of LLM reasoning for complex cases.

We validate the effectiveness of our method on the TinyBench2 benchmark Ye et al. (2024), which surpasses ensemble methods and well-tuned baselines, making further progress on the leaderboard. In summary, our main contributions are as follows:

- We propose a novel tabular context construction method that removes the reliance of LLMs on textual datasets or feature descriptions, thereby significantly enhancing the applicability and privacy-preserving potential of LLMs in tabular domains.
- We present the *Chain of Tabular Thoughts* ($\text{CoT}^2$) approach, which enables step-by-step reasoning and decision-making, effectively unlocking the numerical and logical reasoning capabilities of LLMs on tabular data.
- We are the first to explore the role of LLMs in model ensembling for tabular prediction, addressing a previously overlooked yet crucial component in the modeling pipeline, and extending the use of LLMs beyond existing applications such as feature engineering or data cleaning.

## 2 Related Work

### 2.1 Tabular Data Learning

Tabular data is among the most prevalent data formats in real-world machine learning applications. Gradient-boosted decision trees (GBDTs) Chen & Guestrin (2016); Prokhorenkova et al. (2018); Ke et al. (2017) remain a dominant and highly competitive approach for tabular prediction tasks due to their efficiency and strong empirical performance. As ensemble-based models, GBDTs iteratively construct decision trees to minimize residual loss, making them well-suited for capturing heterogeneous patterns common in tabular datasets Rubachev et al. (2024); Cai & Ye (2025). Meanwhile, the rapid development of deep learning has led to a surge of interest in adapting neural architectures for tabular data Borisov et al. (2022). These efforts include MLP-based variants Klambauer et al. (2017); Gorishniy et al. (2021); Holzmüller et al. (2024), architectures tailored for tabular structures Wang et al. (2017); Chen et al. (2023a), attention-based models Huang et al. (2020); Chen et al. (2023b); Zhou et al. (2023); Jiang et al. (2024), regularization-enhanced frameworks Ye et al. (2023); Wu et al. (2024), and tree-inspired Arik & Pfister (2021); Badirli et al. (2020); Popov et al. (2020); Zhou & Feng (2019) or context-aware methods Gorishniy et al. (2024); Ye et al. (2025b). Despite these innovations, recent large-scale benchmarks Grinsztajn et al. (2022); Ye et al. (2024); McElfresh et al. (2023) consistently show that GBDTs still outperform deep models in most tabular tasks. While several deep learning methods have attempted to mimic ensembling effects Popov et al. (2020); Badirli et al. (2020); Chen et al. (2023c), few have succeeded in consistently closing the gap. Recent advances such as TabM Gorishniy et al. (2025) and BETA Liu & Ye (2025), which integrates BatchEnsemble Wen et al. (2020) into tabular networks, show that efficient and scalable ensembling in deep tabular models remains an active and promising direction.

### 2.2 Language Models for Tabular Prediction

Although Pre-trained Language Models have achieved success in various fields on unseen tasks, their application to tabular data is often limited due to the prevalence of numerical values and the scarcity of textual descriptions. Additionally, concerns over data privacy and security can further restrict the availability of semantic information. As a result, the use of language models in tabular datasets is typically confined to scenarios where textual data is sufficient. TransTab Wang & Sun (2022) trains a tokenizer based on the words present in the tabular data to aid in prediction, rather than using a language model directly. TP-BERTa Yan et al. (2024) does not choose large language models. It fine-tunes relatively smaller pre-trained language models such as RoBERTa Liu et al. (2019) for tabular data prediction. Some other methods start by serializing data through feature names into text, combining this with task descriptions to enable direct predictions by LLMs Dinh et al. (2022); Hegselmann et al. (2023); Gardner et al. (2024). Among them, LIFT Dinh

et al. (2022) requires fine-tuning on the whole training set, while TabuLa-8B Gardner et al. (2024) and TabLLM Hegselmann et al. (2023) focuses on data scarce scenarios.

## 2.3 Retrieval-augmented Generation

Retrieval-Augmented Generation (RAG) was originally developed in the language modeling domain to address the limitations of LLMs on knowledge-intensive tasks Lewis et al. (2020); Gao et al. (2023), enabling models to incorporate external knowledge bases for more accurate and informed responses. However, the use of RAG in tabular data learning remains relatively limited. A notable exception is TabR Gorishniy et al. (2024), which retrieves nearest neighbors to enhance neural tabular model representations. Recent studies such as LocalPFN Thomas et al. (2024) and TabDPT Ma et al. (2024) further demonstrate that leveraging local neighbors to construct context significantly enhances the performance of tabular foundation models (*e.g.* TabPFN Hollmann et al. (2025), TabICL Qu et al. (2025), and TabPTM Ye et al. (2025c)). These approaches suggest that incorporating instance-specific, retrieval-based context not only improves generalization but also facilitates more efficient adaptation to downstream tasks Nagler (2023); Koshil et al. (2024); Ye et al. (2025a). This retrieval-based paradigm has also been extended to enhance tabular prediction with LLMs. Wen et al. (2025) applies the RAG mechanism to enable large language models to effectively process large-scale tabular datasets, constructing informative contexts through instance-level neighbor retrieval. In our approach, we use the labels of retrieved neighbors and the prediction outputs of external models as key components of the context for $CoT^2$'s reasoning. However, instead of relying on the LLM to directly perform classification or regression, we position it as an intelligent ensembling agent. This design allows the LLM to make informed decisions by reasoning over the structured outputs, without accessing any raw tabular features or semantic information. As a result, our method offers strong privacy protection while retaining the benefits of instance-aware, context-driven prediction.

## 2.4 LLMs for Enhancing Machine Learning Pipelines.

Despite the success of machine learning (ML) in real-world tasks, building effective ML pipelines remains challenging due to the many design choices involved. AutoML Hutter et al. (2019) aims to automate this process through methods such as neural architecture search Pham et al. (2018) and Bayesian optimization Frazier (2018). While effective, most AutoML techniques are time-consuming, lack transferability across tasks, and often behave as black boxes with limited interpretability Zhang et al. (2024). To overcome these challenges, recent efforts have explored using Large Language Models (LLMs) to enhance ML workflows. LLM-based agents can assist with various stages of the pipeline—including task understanding Pricope (2025); Chan et al. (2025); Hu et al. (2024), data cleaning Bendinelli et al. (2025); Bodensohn et al. (2025), feature engineering Hollmann et al. (2023b); Nam et al. (2024); Küken et al. (2024); Han et al. (2024); Zhang & Liu (2024), and model building and tuning Li et al. (2024); Zhang et al. (2024; 2023); Huang et al. (2024); Jiang et al. (2025b)—but most of these methods depend heavily on semantic information such as column descriptions or dataset metadata. Notably, no prior work has explored using LLMs as intelligent ensemble experts for tabular prediction tasks. Our approach addresses this gap by treating the LLM not as a direct predictor, but as an instance-aware decision-maker that integrates outputs from multiple external models and nearest-neighbor labels. This enables accurate, interpretable predictions without accessing raw features or semantic cues, thus preserving privacy while enhancing performance.

Our method targets a fundamentally different setting from prior LLM-based approaches for tabular data. Existing methods largely fall into two categories: (1) approaches that convert each instance into a textual prompt using feature names or dataset descriptions, allowing the LLM to act as a predictor Dinh et al. (2022); Hegselmann et al. (2023); Gardner et al. (2024); and (2) LLM-assisted tools that help automate parts of the ML pipeline—such as data cleaning, feature engineering, or hyperparameter tuning—which also rely heavily on task instructions or column-level semantics Bendinelli et al. (2025); Hollmann et al. (2023b); Zhang et al. (2024). In contrast, our method assumes no access to raw features or semantic descriptions. Instead, we position the LLM as an instance-wise ensemble expert that reasons over structured outputs (e.g., model predictions and neighbor labels), enabling accurate and interpretable predictions even in privacy-sensitive or low-semantic settings. Owing to this distinct problem formulation, these existing approaches fall outside the scope of our empirical comparisons.

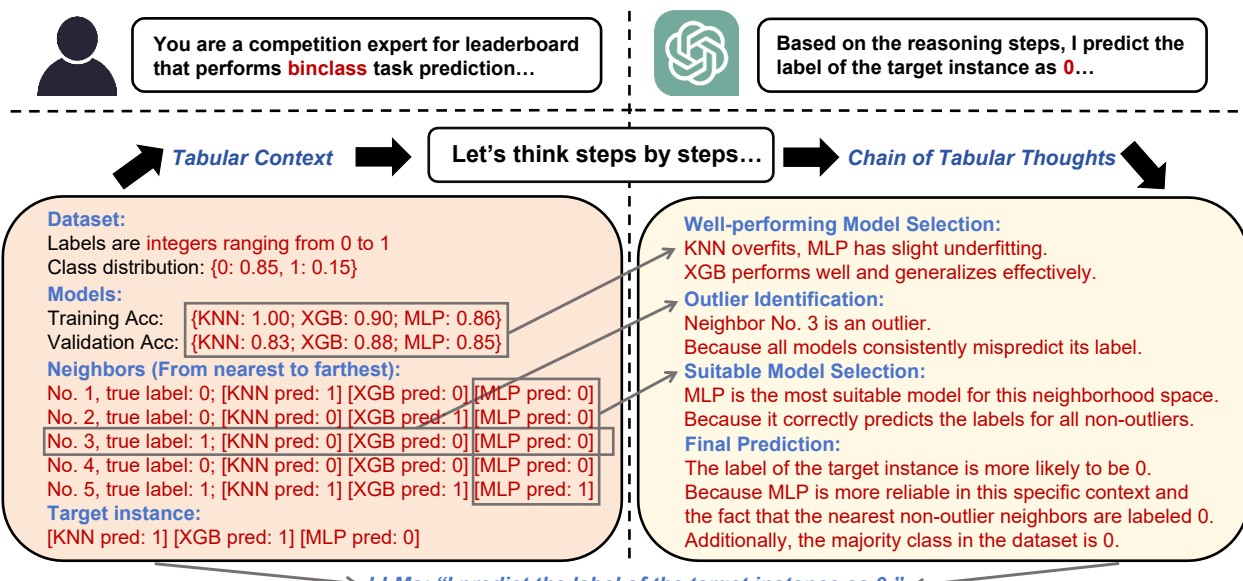

Figure 2: An example of a binary classification task using the tabular context and Chain of Tabular Thoughts (CoT$^2$). We construct the tabular context based on the combination of neighbors and external model predictions. We design reasoning steps by learning from the thought processes of leaderboard experts. Experts typically first filter models and neighbors, then make predictions by aggregating the external models' predictions for the neighbors and target instances. The tabular context and CoT$^2$ are both provided as a prompt to the LLMs. Figure 9 shows an example.

## 3 Methods

Our goal is to leverage LLMs to perform instance-wise ensemble by reasoning over a structured "tabular context," which conveys alternative forms of knowledge without relying on raw features or semantic descriptions. We begin by discussing the background of tabular data learning. Then, we introduce how to create a tabular context around the target instance without textual descriptions. Based on this context, we explain the design of the Chain of Tabular Thought (CoT$^2$), which allows the LLMs to reason clearly.

### 3.1 Preliminary

**Learning with Tabular Data**. Given a labeled tabular dataset $\mathcal{D} = \{(\boldsymbol{x}_i, y_i)\}_{i=1}^{N}$ with $N$ examples (rows in the table). An instance $\boldsymbol{x}_i$ is associated with a label $y_i$. We consider three types of tasks: binary classification $y_i \in \{0, 1\}$, multiclass classification $y_i \in [C] = \{1, \ldots, C\}$, and regression $y_i \in \mathbb{R}$. There are $D$ features (columns) for an instance $\boldsymbol{x}_i$, we denote the $j$-th feature in tabular dataset as $\boldsymbol{x}_{:,j}$ and denote the $j$-th dimension of $\boldsymbol{x}_i$ as $\boldsymbol{x}_{ij}$. We learn multiple tabular models $\mathcal{M} = \{f_m\}_{m=1}^{M}$ on $\mathcal{D}$ that each $f_m$ maps $\boldsymbol{x}_i$ to its label $y_i$. These models exhibit varying generalization capabilities on unseen instances sampled from the same distribution as $\mathcal{D}$. For example, KNN, XGBoost Chen & Guestrin (2016), and Multi-Layer Perceptrons (MLP) are some of the classic models in $\mathcal{M}$.

**Predicting with Large Language Models**. To make predictions on tabular data using LLMs, we need to generate a prompt $p_i$ containing the necessary information based on the target instance $\boldsymbol{x}_i$. Existing methods often construct $p_i$ by utilizing feature descriptions $\{F_i\}_{i=1}^{D}$ and information of dataset $\mathcal{D}$. For example, in TabLLM Hegselmann et al. (2023), $p_i$ includes a textual enumeration of all features. The textual serialization of the $j$-th feature in instance $\boldsymbol{x}_i$ is "The feature name $F_j$ is value $x_{ij}$." The large language model LLM with vocabulary $\mathcal{V}$ generates output text LLM$(p_i) \in \mathcal{V}^*$, which has to be mapped to a valid class in $[C]$ when performing classification. However, when the number of features $D$ is large, the length of the prompt can exceed limitations, and textual descriptions of the dataset may not be available due to data privacy issues or

difficulties associated with data collection. To enable the broad application of LLMs in tabular data, we need prompts that do not rely on textual descriptions.

## 3.2 Tabular Context Based on Neighbors and External Models

To eliminate the limitations imposed by feature descriptions $\{F_i\}_{i=1}^D$ and task descriptions of $\mathcal{D}$, we need to include substitutes for these textual descriptions in the prompt $p_i$. We use re-weighted distance to search for the target instance's nearest neighbors and initially construct a local context. After that, we incorporate predictions from external models into the local context and add other important information to create the final "tabular context."

**Nearest Neighbor Search**. Due to the non-sequential nature of tabular data, tabular data does not have an inherent context. We address this by finding an implicit sequence based on the distance between instances. We calculate the re-weighted distance between the target instance $\boldsymbol{x}_i$ to instance $\boldsymbol{x}_j$ in $\mathcal{D}$:

$$\texttt{dist}(\boldsymbol{x}_i, \boldsymbol{x}_j) = \left(\sum_{l=1}^{D} w_l \cdot |\boldsymbol{x}_{il} - \boldsymbol{x}_{jl}|^d\right)^{\frac{1}{d}}. \tag{1}$$

We set $d = 1$ and $w_l > 0$ is a weight for each dimension. When $w_l = 1$, the distance in Equation 1 degenerates to Manhattan distance ($d = 1$). From the labeled dataset $\mathcal{D}$, we calculate feature weights $w_l$ based on the mutual information (Brown et al., 2012) between features and labels: $w_l = \texttt{norm}\left(\texttt{mutual}(\boldsymbol{x}_{:l}, y)\right)$, where $\texttt{norm}(\cdot)$ normalizes the weights $\{w_l\}_{l=1}^D$ using a min-max scaling method. We rank the distances to obtain the $K$ nearest neighbors $\{\boldsymbol{x}_1, \boldsymbol{x}_2, \ldots, \boldsymbol{x}_K\}$, and their corresponding labels $\{y_1, y_2, \ldots, y_K\}$. The re-weighted distance ensures that neighbors are more similar in important aspects, leading to more meaningful neighbors. The local similarity of neighbors helps provide a relevant and focused context for the target instance. This context can help understand local decision boundaries, leading to more precise and tailored predictions.

**External Models Integration**. External tabular models can provide additional information and compensate for LLMs' numerical reasoning weaknesses. Therefore, we incorporate external models $\mathcal{M} = \{f_m\}_{m=1}^M$ on $\mathcal{D}$ to enrich the context and perform model ensembling. To better apply our method to large datasets, we avoid including feature values in the context, as this would inevitably constrain the prompt length. The knowledge between feature values and labels learned by the external models helps mitigate this information loss. An expert can more accurately infer the most suitable external models for the target instance by analyzing the relationship between the neighbors' true labels and the model predictions. Consequently, we combine the capabilities of trained traditional tabular models with the in-context learning abilities of LLMs. Based on neighbors and external models, the tabular context in our designed prompt $p_i = \texttt{context}\left(\{y_j\}_{j=1}^N, \{(\boldsymbol{x}_j, y_j)\}_{j=1}^K, \mathcal{M}\right)$ includes:

- The basic meta information of dataset $\mathcal{D}$, such as the label set $[C]$ in classification and the label range in regression. In classification tasks, we include the label frequencies $\{q_i\}_{i=1}^C$ in $\mathcal{D}$, where $q_i = \sum_{j=1}^N \mathbb{I}(y_j = i)/N$ and $\mathbb{I}(\cdot)$ is the indicator function.
- The training accuracies $\{\texttt{train\_acc}(f_m)\}_{m=1}^M$ and validation accuracies $\{\texttt{val\_acc}(f_m)\}_{m=1}^M$ of each model. These elements are already saved during the construction of $\mathcal{M}$, and both the training and validation sets come from the partitioning of dataset $\mathcal{D}$, without introducing additional data.
- The true labels of these neighbors $\{y_j\}_{j=1}^K$, and the predictions of $M$ external models for these neighbors $\{\{f_m(\boldsymbol{x}_j)\}_{m=1}^M\}_{j=1}^K$. These elements can be obtained through $\{(\boldsymbol{x}_j, y_j)\}_{j=1}^K$ and $\mathcal{M}$.
- The external models' predictions for target instance $\{f_m(\boldsymbol{x}_i)\}_{m=1}^M$.

Without including semantic content, we have constructed a tabular context rich in information within the prompt. We anticipate that the robust expert knowledge of LLMs will be able to synthesize this evidence and carry out instance-wise model integration for target instance $x_i$:

$$\hat{y}_i = \texttt{map}\left(\texttt{LLM}(p_i)\right) = \texttt{map}\left(\texttt{LLM}\left(\texttt{context}\left(\{y_j\}_{j=1}^N, \{(\boldsymbol{x}_j, y_j)\}_{j=1}^K, \mathcal{M}\right)\right)\right), \tag{2}$$

where $\texttt{map}(\cdot)$ extracts the final prediction from the LLM's response through regular expression matching. For example, in classification tasks, we inform LLMs that we will extract the label from their response using the following code:

```
label = re.search(r'I predict the label of the target instance as (\d+)',
                  your_response_text).group(1)
```

### 3.3 Chain of Tabular Thoughts Based on Tabular Context

LLMs often struggle with multi-step or complex reasoning tasks. Our experiments in Table 1 find that it is challenging for LLMs to directly derive accurate answers from our tabular context. The CoT helps by breaking down the problem into smaller tasks, allowing the model to focus on each step individually. Therefore, we emulate an expert's analysis on the leaderboard and add some reasoning steps to prompt $p_i$. We design the Chain of Tabular Thoughts ($\text{CoT}^2$) to help LLMs reason within our tabular context. Take classification for example, our reasoning steps are as follows:

a) **Well-performing Model Selection**. Based on the training accuracies $\{\texttt{train\_acc}(f_m)\}_{m=1}^{M}$ and validation accuracies $\{\texttt{val\_acc}(f_m)\}_{m=1}^{M}$ of each model, LLMs infer the overall performance of the external models on the dataset. Then LLMs select $M^w$ well-performing models $\{f_m\}_{m=1}^{M^w}$ from external models:

$$\{f_m\}_{m=1}^{M^w} = \texttt{step\_a}\left(\{\texttt{train\_acc}(f_m)\}_{m=1}^{M}, \{\texttt{val\_acc}(f_m)\}_{m=1}^{M}\right). \tag{3}$$

We aim for LLMs to identify overfitting and underfitting models based on their training and validation accuracies, and to find the overall well-performing models on $\mathcal{D}$.

b) **Outlier Identification**. Based on the true labels of the neighbors $\{y_i\}_{i=1}^{K}$, the neighbors' predicted labels from well-performing models $\{\{f_m(\boldsymbol{x}_j)\}_{m=1}^{M^w}\}_{j=1}^{K}$, and the label frequencies $\{q_i\}_{i=1}^{C}$, LLMs identify non-outliers $\{y_j\}_{j=1}^{K^*}$ among the neighbors. Here, we use the true labels to refer to the neighbors:

$$\{y_j\}_{j=1}^{K^*} = \texttt{step\_b}\left(\{y_i\}_{i=1}^{K}, \{\{f_m(\boldsymbol{x}_j)\}_{m=1}^{M^w}\}_{j=1}^{K}, \{q_i\}_{i=1}^{C}\right). \tag{4}$$

If the majority of well-performing models predict incorrectly for a particular neighbor, it suggests that this neighbor might be an outlier, negatively affecting the predictions. We want the LLMs to be able to identify such outliers. Label frequencies provide additional information about the degree of data imbalance, which aids in reasoning.

c) **Suitable Model Selection**. Based on the true labels of the non-outliers $\{y_j\}_{j=1}^{K^*}$, the non-outliers' predicted labels from all models $\{\{f_m(\boldsymbol{x}_j)\}_{m=1}^{M}\}_{j=1}^{K^*}$, and the label frequencies $\{q_i\}_{i=1}^{C}$, LLMs select $M^s$ the most suitable models $\{f_m\}_{m=1}^{M^s}$ for the neighborhood space of the target instance:

$$\{f_m\}_{m=1}^{M^s} = \texttt{step\_c}\left(\{y_j\}_{j=1}^{K^*}, \{\{f_m(\boldsymbol{x}_j)\}_{m=1}^{M}\}_{j=1}^{K^*}, \{q_i\}_{i=1}^{C}\right). \tag{5}$$

Models that perform well overall on the dataset may not be the most efficient at predicting the target instance. It is essential to identify the best-suited models for the target instance within the neighbor space after filtering out outliers.

d) **Final Prediction**. Based on the true labels of the non-outliers $\{y_j\}_{j=1}^{K^*}$, the label frequencies $\{q_i\}_{i=1}^{C}$, and the target instance $\boldsymbol{x}_i$'s predicted labels from the most suitable models and well-performing models $\{f_m(\boldsymbol{x}_i)\}_{m=1}^{M^s} \cup \{f_m(\boldsymbol{x}_i)\}_{m=1}^{M^w}$, LLMs make the prediction for the target instance's label $\hat{y}_i$:

$$\hat{y}_i = \texttt{step\_d}\left(\{y_j\}_{j=1}^{K^*}, \{f_m(\boldsymbol{x}_i)\}_{m=1}^{M^s} \cup \{f_m(\boldsymbol{x}_i)\}_{m=1}^{M^w}, \{q_i\}_{i=1}^{C}\right). \tag{6}$$

After removing outliers and unsuitable external models, LLMs can use a KNN-based approach and model ensembling within the clean local context to achieve the most confident final predictions. Well-performing models, being the strongest models on the current dataset leaderboard, provide auxiliary information for the final prediction.

Finally, we summarize the reasoning steps into text $t$ and include them in prompt $p_i$ in Equation 2. The tabular context and the Chain of Tabular Thoughts are combined into the final prompt $\tilde{p}_i$, which is then input into the LLMs to obtain the final prediction:

$$\hat{y}_i = \texttt{map}\left(\texttt{LLM}(\tilde{p}_i)\right) = \texttt{map}\left(\texttt{LLM}\left(p_i \cup t\right)\right). \tag{7}$$

**Variant for Regression Tasks**. For the regression task, we remove the label frequency, retain the true labels and model predictions to four decimal places. We use RMSE instead of accuracy. The regular expression for map$(\cdot)$ was changed to `(-?d+.d+)`. Figure 10 shows an example of the prompts. If the match fails, we will re-enter the prompt until it succeeds. During our experiments, there was no instance of consecutive matching failures occurring 10 times.

**A Simple Alternative Approach.** To assess the necessity of introducing large language models, we design a non-LLM baseline named **MetaXGB**, which utilizes the same components as our constructed tabular context. For each target instance $\boldsymbol{x}_i$ in the validation or test set, we retrieve its $K$ nearest neighbors from the training set using the re-weighted distance in Equation 1, and collect their true labels as well as the external model predictions on both the target and its neighbors. These components are concatenated into a fixed-length feature vector:

$$\mathbf{z}_i = \left[ \{f_m(\boldsymbol{x}_i)\}_{m=1}^M, \{y_j\}_{j=1}^K, \{\{f_m(\boldsymbol{x}_j)\}_{m=1}^M\}_{j=1}^K \right], \tag{8}$$

which is then used to train a downstream XGBoost classifier on the validation set. The trained model is evaluated on the test set, and results are compared with our proposed method in subsection 4.2.

**Hard Sample Identification.** As discussed in section 1, a major challenge in deploying LLM-based tabular methods is their high inference cost, as the LLM must be invoked for each instance. However, in many real-world scenarios, most samples can already be accurately predicted by multiple external models with high agreement. To reduce computational overhead, we adopt a selective strategy that reserves LLM reasoning for more difficult cases—those where external models disagree. Taking classification tasks as an example, we define a *hard sample* as one for which fewer than a fraction $\tau$ of the $M$ external models predict the same class label. In other words, if more than $\tau$ of the models agree on the prediction, the instance is considered *easy*, and LLM inference can be skipped.

**Summary**. To address the three challenges of applying LLMs to tabular data, $\text{CoT}^2$ introduces the following solutions:

- $\text{CoT}^2$ designs an information-rich tabular context to replace textual descriptions, freeing LLMs from relying on dataset semantics.
- $\text{CoT}^2$ helps LLMs leverage the capabilities of external models to understand the numerical relationships between features and labels. Additionally, clear reasoning steps are included to assist LLMs in understanding the relationship between model predictions, neighbor labels, and target predictions.
- To reduce inference cost and avoid token limits from including raw features, $\text{CoT}^2$ adopts a selective strategy: LLMs are only invoked for hard instances where external models disagree, while easy cases are handled without LLM reasoning.

# 4 Experiments

## 4.1 Setups

**Datasets**. To evaluate the effectiveness of $\text{CoT}^2$ on challenging tabular prediction tasks, we adopt the TinyBench2 Benchmark Suite Ye et al. (2024), a representative subset of 45 datasets selected from a larger benchmark containing over 300 datasets. The full benchmark is designed for evaluating tabular models across diverse data types and task settings. However, due to its scale, it poses a high computational burden for model evaluation. TinyBench2 addresses this challenge by selecting 15% of datasets while preserving the relative ranking of models. The selection process is framed as an optimization problem: minimizing the mean absolute error (MAE) between average model ranks on the subset and the full benchmark. The final TinyBench2 shows the best consistency on both seen and unseen models. By using TinyBench2, we efficiently evaluate our method while ensuring the results are representative of full-scale benchmarks Ye et al. (2024).

***Remark***. $\text{CoT}^2$ does not require providing dataset descriptions or raw feature values as input to the LLM. Instead, the LLM context is constructed solely from the predictions of external models on the target test sample and the labels and predictions of its nearest neighbors. As a result, we do not need to consider potential dataset leakage during LLM pretraining, nor do we require dedicated dataset leakage detection

procedures when selecting evaluation datasets. This makes our approach more broadly applicable, especially when using proprietary or privacy-sensitive tabular data Bordt et al. (2024); Küken et al. (2024).

**Model Set Selection**. To ensure a comprehensive and robust evaluation of ensemble performance, we construct a model set that spans multiple paradigms of tabular modeling. Our goal is twofold: to cover the dominant families of models used in practice, and to expose the ensemble mechanism to diverse inductive biases. Specifically, we include:

a) **Three representative GBDT models**: XGBoost Chen & Guestrin (2016), LightGBM Ke et al. (2017), and CatBoost Prokhorenkova et al. (2018), which are widely recognized as state-of-the-art models for tabular data due to their strong performance, robustness, and widespread adoption in both academia and industry.
b) **Four deep learning models for tabular data**: MLP, ResNet, and FT-Transformer Gorishniy et al. (2021), which are representative architectures selected by Ye et al. (2024) based on systematic benchmarking. To broaden architectural diversity, we also include AutoInt Song et al. (2019), a hybrid model bridging tabular deep learning and recommender systems that integrates attention mechanisms and feature interaction modeling.
c) **A classical non-parametric method**: K-Nearest Neighbors (KNN), which provides an intuitive, instance-based learning paradigm. Including KNN complements the parametric models and offers a contrasting local inductive bias that is useful for diversity in ensemble behavior.

All models are trained independently on each dataset, and their predictions are used by our method and the baselines to construct tabular contexts and evaluate ensemble performance. This carefully chosen model set balances accuracy, architectural diversity, and modeling philosophy.

**Comparison Methods.** We compare two main categories of methods, both derived from a common set of tabular models that also serve as the external model pool for $CoT^2$:

- **TinyBench2 Baseline Methods**: This category includes all baseline methods reported in the TinyBench2 benchmark Ye et al. (2024), which already cover all the models in our model set. These include classical machine learning models, gradient boosted decision trees (GBDTs), and deep learning architectures for tabular data. In addition, we also compare against TabM Gorishniy et al. (2025), a recently proposed deep ensemble learning method that achieves strong performance.
- **Ensemble Methods over the Model Set**: Based on the same model set, we implement several standard ensemble or selection strategies for comparison:
  - Best Model: selects the model with the highest validation accuracy on each dataset;
  - Average Voting: averages the predicted logits across models;
  - Weighted Voting: averages logits weighted by each model's training accuracy.
- **Non-LLM Context-based Baseline (MetaXGB)**: We further compare with **MetaXGB** (see subsection 3.3), a simple non-LLM baseline using the same tabular context with $CoT^2$.

**Evaluation Protocol.** We follow the evaluation protocol proposed in Ye et al. (2024) to ensure fair and consistent comparisons across all methods. Specifically, we randomly split each dataset into training, validation, and test sets with a ratio of 64% : 16% : 20%. The validation set is used for model selection and early stopping where applicable. All methods, including those in our model set and all comparison baselines, are trained and evaluated on the same data splits. To account for randomness, we repeat each experiment five times with different random seeds {0, 1, 2, 3, 4} and report the average performance on the test set. For classification tasks, we report average accuracy (Acc), and for regression tasks, we report average Root Mean Squared Error (RMSE).

### 4.2  Results

**Performance on Standard Tasks**. For $CoT^2$, we use `gpt-3.5-turbo` and `Deepseek-v3` DeepSeek-AI et al. (2024) with a temperature setting of 0.2. We set the number of neighbors to 10. The external models used are shown in subsection 4.1. As shown in Figure 3, Our method achieves the best average ranking across all classification datasets. For regression tasks, the performance results are provided in Appendix C.

Table 1: Mean and STD of test accuracy on five datasets. $CoT^2$ provided significant improvements for GPT-3.5 and smaller benefits for GPT-4o and Deepseek-v3, indicating that the reasoning steps in $CoT^2$ align well with advanced expert knowledge. (**Bold** indicates superiority across all methods, while underline signifies whether $CoT^2$ has brought improvements to the same LLM.)

| Dataset | gpt-3.5-turbo | | gpt-4o | | Deepseek-V3 | |
|---|---|---|---|---|---|---|
| | w/o $CoT^2$ | w/ $CoT^2$ | w/o $CoT^2$ | w/ $CoT^2$ | w/o $CoT^2$ | w/ $CoT^2$ |
| BAS | $93.36 \pm 0.28$ | $\mathbf{94.63} \pm 0.18$ | $94.40 \pm 0.00$ | $\mathbf{94.63} \pm 0.18$ | $94.10 \pm 0.28$ | $\underline{94.55} \pm 0.18$ |
| DIS | $98.41 \pm 0.05$ | $\underline{98.54} \pm 0.00$ | $98.57 \pm 0.05$ | $\underline{98.60} \pm 0.06$ | $98.54 \pm 0.15$ | $\mathbf{98.68} \pm 0.06$ |
| SYL | $91.65 \pm 0.18$ | $\underline{94.56} \pm 0.04$ | $94.60 \pm 0.30$ | $\underline{94.87} \pm 0.24$ | $94.43 \pm 0.27$ | $\mathbf{94.93} \pm 0.14$ |
| CRE | $76.37 \pm 0.25$ | $\underline{77.86} \pm 0.08$ | $77.99 \pm 0.15$ | $\mathbf{78.01} \pm 0.15$ | $77.80 \pm 0.15$ | $\underline{77.97} \pm 0.16$ |
| FOR | $64.18 \pm 0.26$ | $\underline{68.65} \pm 0.11$ | $69.66 \pm 0.19$ | $\mathbf{70.69} \pm 0.11$ | $69.75 \pm 0.27$ | $\mathbf{70.69} \pm 0.11$ |
| Mean | 84.79 | 86.85 | 87.04 | **87.36** | 86.92 | **87.36** |

**Effective of $CoT^2$**. As shown in Figure 3, $CoT^2$ significantly outperforms the non-LLM baseline MetaXGB, which adopts a hard-rule strategy based on handcrafted feature construction and a downstream XGBoost classifier. This result highlights the limitations of rigid integration methods and demonstrates the necessity of leveraging large language models for more intelligent and flexible model ensembling. The key difference between using and not using the chain of tabular thoughts is whether the four inference steps are included in the prompt. Incorporating $CoT^2$ significantly enhances performance when using GPT-3.5 compared to the original tabular context, as shown in Table 1. We further include comparisons with `gpt-4o`, showing that $CoT^2$ continues to bring benefits for more capable models. Figure 11 and Figure 12 show that, without $CoT^2$, GPT-3.5's predictions rely solely on the models that perform well on the overall dataset and majority prediction, resulting in an incorrect prediction. $CoT^2$ enables GPT-3.5 to perform clear and structured reasoning in the tabular context, leading to a correct prediction.

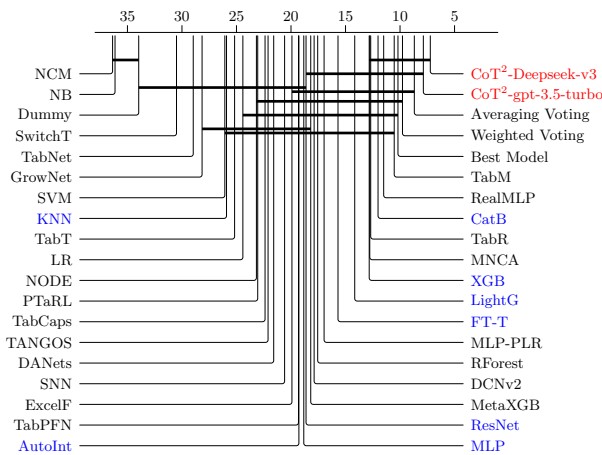

Figure 3: Critical difference diagram based on the Wilcoxon-Holm test with a significance level of 0.05, used to assess pairwise significance of methods on 30 classification datasets in TinyBench2. Blue-colored methods represent the models included in the external model set. The method names in the diagram are abbreviated; the mapping from abbreviations to full names can be found in Ye et al. (2024) and Appendix A.

The effectiveness of $CoT^2$ helps bridge the performance gap between GPT-3.5 and GPT-4o in this specific reasoning task, demonstrating that our designed reasoning steps align with the more advanced expert knowledge in GPT-4o. With $CoT^2$, our simple and efficient prediction context does not require new or complex knowledge. The responses of different LLMs to the same prompt are shown in Figure 13, and Figure 14. We also include responses from the latest version of `ChatGPT` in Figure 15, Figure 16, Figure 17, and Figure 18. We observe that both `gpt-4o` and `Deepseek-v3` tend to provide more fine-grained analysis for each piece of information. In particular, `Deepseek-v3` and the latest ChatGPT often structure their reasoning in a list format, which enhances interpretability and clarity.

**Reducing Inference Cost via Selective LLM Usage**. As discussed in section 1, a key challenge of LLM-based tabular prediction is the high inference cost, as separate prompts must be processed for each instance Dinh et al. (2022); Hegselmann et al. (2023); Gardner et al. (2024). To reduce inference cost, we adopt a strategy to identify easy instances—those for which external models show high agreement. Specifically, for classification tasks, we define an instance as easy if at least $\tau = 3/4$ of the external models agree on the prediction, and LLM inference is skipped in these cases. This selective strategy significantly reduces

computational overhead by reserving LLM inference for more challenging instances. As shown in Table 2, this approach allows us to bypass LLM reasoning for the majority of test samples, ensuring that LLMs are used only when their reasoning capabilities are most needed.

### 4.3 Ablation Study

To better understand the design choices in $CoT^2$, we conduct an ablation study on several key components. All the ablation experiments are conducted using the `gpt-3.5-turbo` model on five classification datasets in TinyBench2, and the results are reported in Appendix B.

**Model Set**: We investigate the impact of the external model set on the performance of $CoT^2$ by varying both the number and quality of models included. Specifically, we experiment with different pool sizes and progressively introduce stronger models into the ensemble. We evaluate three configurations: a reduced model set of 4 strong models (XGBoost, CatBoost, MLP, FT-Transformer), the original 8-model pool used in the main experiments, and an extended 12-model set that adds four recent, higher-performing deep models (RealMLP Holzmüller et al. (2024), TabR Gorishniy et al. (2024), ModernNCA Ye et al. (2025b), and TabM Gorishniy et al. (2025)). The results demonstrate that increasing the number of models generally enhances performance, while incorporating higher-performing models into the pool leads to further gains, as shown in Figure 4.

**Number of Neighbors ($k$)**: We evaluate different values of $k$ when constructing the tabular context. The results show that moderate values (*e.g.*, $k = 10$) strike a good balance between context richness and prompt length, as shown in Figure 5.

Additionally, we summarize key design choices and ablation factors affecting the performance of $CoT^2$, including distance metrics, model name anonymization, LLM inference temperature, and the hard-sample selection threshold. Detailed results and discussions are provided in Appendix B.

## 5 Conclusion

The widespread use of LLMs on tabular data is limited by several factors: a heavy reliance on textual descriptions, an inability to handle datasets with a large number of features, and insensitivity to numerical values. To apply the expert knowledge of LLMs to aid in predictions on tabular data, we designed a tabular context incorporating instance-specific insights as a substitute for semantic descriptions and feature values. By utilizing the capabilities of external models, we addressed the weaknesses of LLMs in handling the relationship between numerical features and labels. Additionally, we devised a chain of tabular thoughts to teach LLMs how to comprehend numerical values within our tabular context. Our method can be efficiently applied to tabular prediction tasks.

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

The Appendix consists of four sections:

- Appendix A: We provide detailed descriptions of the datasets used in our experiments, along with implementation details for reproducibility.
- Appendix B: We present a comprehensive ablation study analyzing the impact of key design choices in our method.
- Appendix C: We include complete experimental results that were omitted from the main paper due to space limitations.
- Appendix D: We show representative examples of our method, including prompt formats and responses under different settings.

## A    Datasets and implementation details

To facilitate comprehensive evaluation and analysis, we adopt the TinyBench2 benchmark Ye et al. (2024), which includes a diverse collection of 45 tabular datasets spanning different task types, including binary classification, multiclass classification, and regression. These datasets vary widely in size, feature composition, and difficulty, making them suitable for robust and fair assessment of tabular learning methods.

Table 2 summarizes the key statistics for each dataset used in our experiments. Specifically, we report the following information:

- **Abbr**: A short identifier used throughout the paper for concise reference.
- **Task_type**: The type of machine learning task (regression, binclass, or multiclass).
- **N / C**: The number of numerical and categorical features, respectively.
- **Samples**: The total number of instances in the dataset.
- **Hard ratio**: The percentage of hard samples, indicating the dataset's learning difficulty.

We quantify dataset difficulty using the **hard ratio**, which represents the proportion of hard samples in each dataset. A sample is considered *hard* if it fails to reach consensus among external models during evaluation. For classification tasks, a sample is labeled as hard if fewer than 3/4 of the external models predict the same class label. For regression tasks, we use an outlier-based rule: a sample is marked as hard if more than 1/4 of the external model predictions fall outside the interquartile range (IQR), specifically beyond $[Q_1 - 1.5 \times \mathrm{IQR}, Q_3 + 1.5 \times \mathrm{IQR}]$. These criteria help identify instances that are difficult to predict consistently, providing a measure of dataset complexity.

**External models.** For all external baseline models that do not explicitly specify preprocessing strategies for categorical and numerical features—such as MLP and ResNet—we uniformly apply one-hot encoding for categorical features and standard normalization for numerical features and regression labels. Training is performed with a maximum of 200 epochs, a batch size of 1024, and early stopping with a patience of 20 epochs. We conduct 100 rounds of hyperparameter tuning for each external model Liu et al. (2024). The full search space configurations are available at `https://github.com/LAMDA-Tabular/TALENT/tree/main/TALENT/configs/opt_space`.

**CoT$^2$ Configuration.** In our main experiments, we run each dataset using 5 different random seeds and report the average accuracy (for classification) or RMSE (for regression). For each target instance, we retrieve $k = 10$ nearest neighbors from the training set as context, and set the temperature parameter to $t = 0.2$. We deploy the CoT$^2$ pipeline using two large language models: `gpt-3.5-turbo-0125` and `DeepSeek-V3-P001`. Additionally, results from `gpt-4o` are reported in Table 1 for further comparison.

**Abbreviations of models compared in our main experiments.** We group all baseline methods into several categories for clarity. Classical methods include Dummy, Logistic Regression (LR), K-Nearest Neighbors (KNN), Support Vector Machines (SVM), Naive Bayes, Linear Regression (LR), and DNNR. Tree-based methods include Random Forest (RF), XGBoost (XGB) Chen & Guestrin (2016), LightGBM (LightG) Ke et al. (2017), and CatBoost (CatB) Prokhorenkova et al. (2018). MLP variants cover vanilla MLP, MLP-PLR Gorishniy et al. (2022), Self-Normalizing Neural Networks (SNN) Klambauer et al. (2017), ResNet Gorishniy et al. (2021), RealMLP Holzmüller et al. (2024), and TabM Gorishniy et al. (2025). Special architectures include DCNv2 Wang et al. (2021), DANets Chen et al. (2022), and TabCaps Chen et al.

Table 2: The list of datasets in TinyBench2 Ye et al. (2024), along with the statistics for each dataset.

| Dataset | Abbr | Task_type | N | C | Samples | Hard ratio |
|---|---|---|---|---|---|---|
| Ailerons | AIL | regression | 40 | 0 | 13750 | 49.3818 |
| BNG(breast-w) | BWR | binclass | 9 | 0 | 39366 | 0.5588 |
| BNG(cmc) | CMC | multiclass | 2 | 7 | 55296 | 9.5931 |
| BNG(tic-tac-toe) | TTT | binclass | 0 | 9 | 39366 | 4.6355 |
| CPMP-2015-regression | C2R | regression | 23 | 2 | 2108 | 53.7915 |
| Cardiovascular-Disease-dataset | CDD | binclass | 5 | 6 | 70000 | 3.9214 |
| CookbookReviews | COO | regression | 7 | 0 | 18182 | 4.1793 |
| FOREX_audchf-day-High | ADH | binclass | 10 | 0 | 1833 | 28.0654 |
| FOREX_audsgd-hour-High | AHH | binclass | 10 | 0 | 43825 | 26.5830 |
| FOREX_cadjpy-hour-High | FOR | binclass | 10 | 0 | 43825 | 21.5402 |
| Gender_Gap_in_Spanish_WP | GGI | multiclass | 13 | 0 | 4746 | 10.5263 |
| IEEE80211aa-GATS | IGE | regression | 27 | 0 | 4046 | 46.7901 |
| KDD | KDD | binclass | 34 | 11 | 5032 | 12.1152 |
| Large-scale_Wave_Energy_Farm_Sydney_49 | LSW | regression | 99 | 0 | 17964 | 43.6961 |
| Superconductivty | SUP | regression | 81 | 0 | 21197 | 37.9953 |
| VulNoneVul | VUL | binclass | 16 | 0 | 5692 | 0.0000 |
| archive2 | ARC | regression | 11 | 1 | 1143 | 34.0611 |
| bank8FM | BAN | regression | 8 | 0 | 8192 | 64.2465 |
| baseball | BAS | multiclass | 15 | 1 | 1340 | 1.4925 |
| communities_and_crime | CAC | regression | 102 | 0 | 1994 | 36.3409 |
| credit | CRE | binclass | 10 | 0 | 16714 | 10.5594 |
| dis | DIS | binclass | 6 | 23 | 3772 | 0.3974 |
| eye_movements_bin | EMB | binclass | 20 | 0 | 7608 | 25.9527 |
| fried | FRI | regression | 10 | 0 | 40768 | 71.4251 |
| healthcare_insurance_expenses | HIE | regression | 3 | 3 | 1338 | 27.9851 |
| house_16H_reg | H1R | regression | 16 | 0 | 22784 | 32.1922 |
| jungle_chess_2pcs_raw_endgame_complete | JC2 | multiclass | 6 | 0 | 44819 | 12.0259 |
| kin8nm | KIN | regression | 8 | 0 | 8192 | 33.3130 |
| law-school-admission-bianry | LSA | binclass | 7 | 4 | 20800 | 0.0000 |
| mfeat-fourier | MFF | multiclass | 76 | 0 | 2000 | 11.0000 |
| mv | MV | regression | 7 | 3 | 40768 | 91.2558 |
| online_shoppers | OSN | binclass | 5 | 9 | 12330 | 4.9067 |
| page-blocks | PBA | multiclass | 10 | 0 | 5473 | 1.4612 |
| pc3 | PC3 | binclass | 37 | 0 | 1563 | 3.8339 |
| pendigits | PEN | multiclass | 16 | 0 | 10992 | 0.5457 |
| qsar_fish_toxicity | QFT | regression | 4 | 2 | 908 | 31.3187 |
| rl | RL | binclass | 5 | 7 | 4970 | 24.3461 |
| satimage | SAT | multiclass | 36 | 0 | 6430 | 5.5210 |
| segment | SEG | multiclass | 17 | 0 | 2310 | 6.2771 |
| sylvine | SYL | binclass | 20 | 0 | 5124 | 3.3171 |
| taiwanese_bankruptcy_prediction | TBP | binclass | 95 | 0 | 6819 | 0.8798 |
| waveform-5000 | W5A | multiclass | 40 | 0 | 5000 | 6.7000 |
| website_phishing | WPE | multiclass | 0 | 9 | 1353 | 8.1181 |
| wine-quality-white | WQW | multiclass | 11 | 0 | 4898 | 24.5918 |
| yeast | YEA | multiclass | 8 | 0 | 1484 | 16.1616 |

(2023a). Token-based methods include AutoInt Song et al. (2019), TabTransformer (TabT) Huang et al. (2020), FT-Transformer (FT-T) Gorishniy et al. (2021), and ExcelFormer (ExcelF) Chen et al. (2023b). Regularization-based methods comprise TANGOS Jeffares et al. (2023), SwitchTab (SwitchT) Wu et al. (2024), and PTaRL Ye et al. (2023). Tree-mimic methods include NODE Popov et al. (2020), GrowNet Badirli et al. (2020), and TabNet Arik & Pfister (2021). Context-based methods include TabR Gorishniy et al. (2024), TabPFN Hollmann et al. (2023a) and ModernNCA (MNCA) Ye et al. (2025b).

# B   Ablation Study

The additonal ablation experiments, as shown in Figure 6, Figure 7, Figure 8, and Table 3, are conducted on a subset of five datasets: **BAS**, **DIS**, **SYL**, **CRE**, and **FOR**.

- **Distance Metric**: We compare several distance metrics for neighbor retrieval, including Manhattan, Euclidean, cosine similarity, and our proposed re-weighted distance in Equation 1. These metrics affect how relevant neighbors are selected for each target instance, which in turn influences the quality of the constructed tabular context (Figure 6).
- **Anonymizing External Model Names**: We examine whether hiding the real names of external models in the tabular context affects CoT$^2$'s performance. Instead of using actual model names, we substitute them with anonymized labels (*e.g.*, Model A, B, C, D). Interestingly, we observe improved performance on four out of five datasets under this anonymized setting. This suggests that LLMs may carry inherent biases or preferences toward certain model names, and removing these cues can lead to more objective and consistent reasoning (Figure 7).
- **LLM Inference Temperature**: We analyze the effect of temperature settings on model outputs. Lower temperatures (*e.g.*, 0.2) yield more stable and deterministic predictions, while higher temperatures introduce variability and may reduce accuracy (Figure 8).
- **Threshold for Hard Sample Selection**: We study how varying the agreement threshold for identifying hard samples affects both predictive performance and inference cost (Table 3).

Table 3: Effect of varying the hard sample threshold on accuracy and inference cost. Increasing the threshold allows more instances to be handled by the LLM, but may decrease accuracy due to potential hallucinations on simple instances. Conversely, decreasing the threshold may also reduce accuracy, as difficult samples not solvable by base ensembles alone may be excluded from LLM inference. Results are reported on the SYL and CRE datasets.

| Dataset | Threshold | Accuracy (%) | Time (s) | Tokens (input) | Tokens (output) | Price ($) |
|---------|-----------|--------------|----------|----------------|-----------------|-----------|
|         | 0.50      | 94.20        | 19.3     | 41715          | 6148            | 0.03      |
| SYL     | 0.75      | 94.56        | 62.4     | 74920          | 11841           | 0.06      |
|         | 1.00      | 93.06        | 213.2    | 394769         | 60807           | 0.29      |
|         | 0.50      | 77.08        | 146.3    | 275757         | 41475           | 0.20      |
| CRE     | 0.75      | 77.86        | 644.8    | 1525206        | 134162          | 0.96      |
|         | 1.00      | 77.29        | 1418.4   | 2995927        | 454875          | 2.18      |

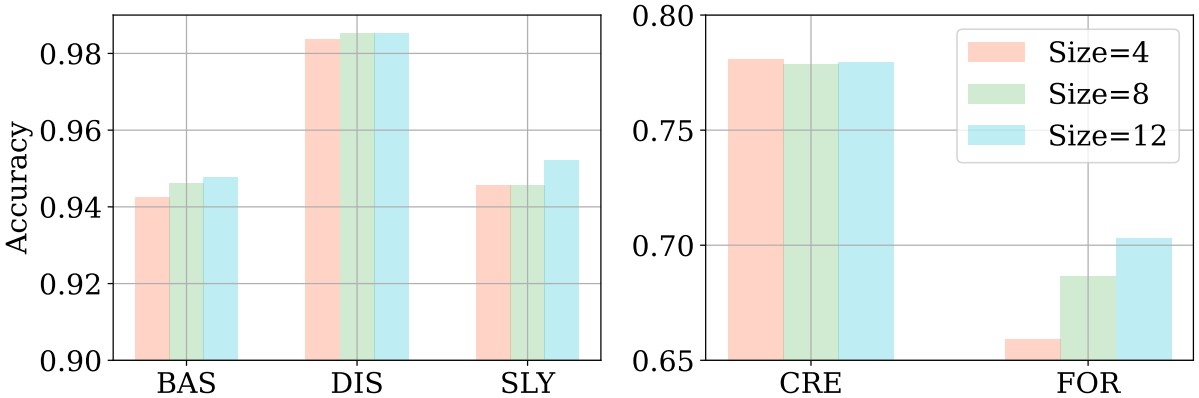

Figure 4: Impact of external model set size and quality on the performance of CoT$^2$.

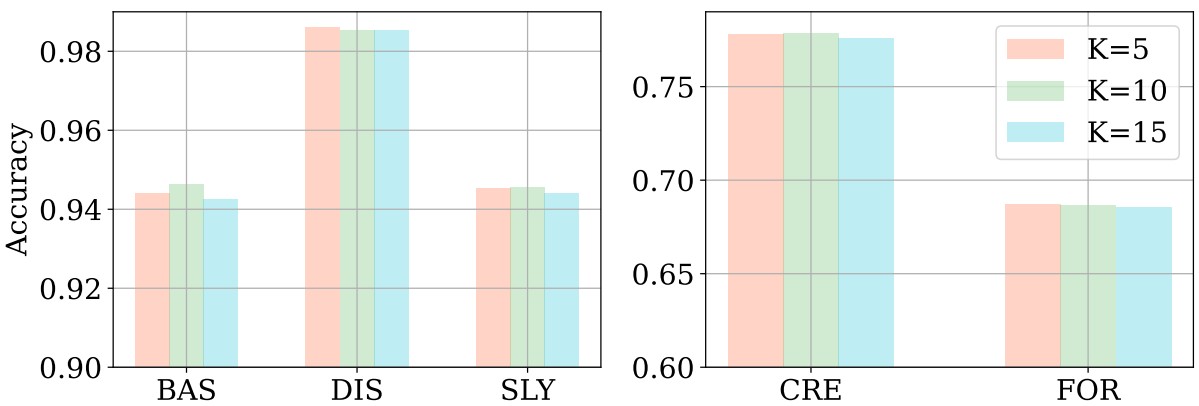

Figure 5: Performance of CoT$^2$ under different numbers of neighbors $k$ used in the context.

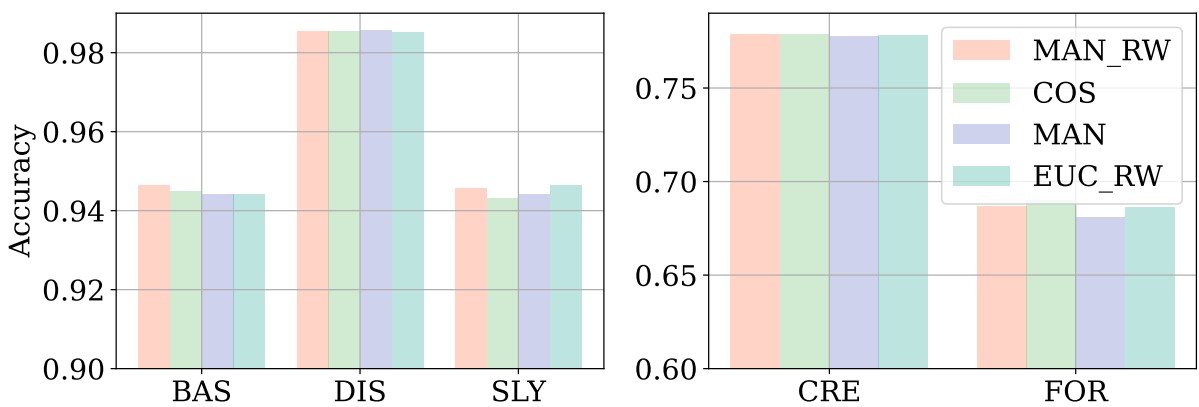

Figure 6: In the process of nearest neighbor search, we used the Manhattan distance reweighted by mutual information (MAN-RW) in the main experiment. We also experimented with cosine distance (COS) and Euclidean distance reweighted by mutual information (EUC-RW). The knowledge from LLMs and the predictions from external models can help us filter out outliers in the nearest neighbors, making CoT$^2$ robust to different distance metrics.

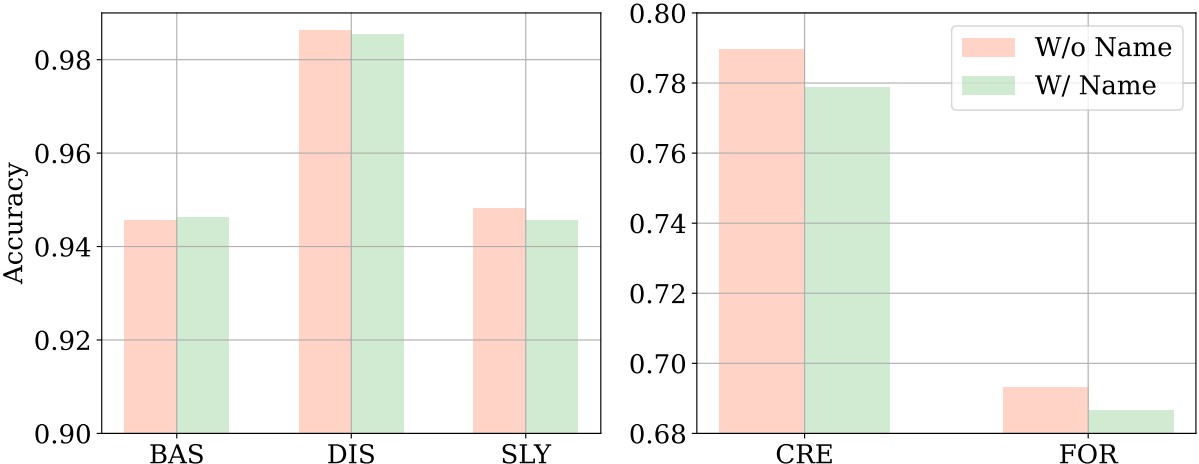

Figure 7: Impact of anonymizing external model names in the tabular context on CoT²'s performance. We compare two settings: *w/ name*, where real model names are provided, and *w/o name*, where anonymized labels (*e.g.*, Model A, B, C) are used. Results show that the anonymized version (*w/o name*) outperforms the named version on four out of five datasets, indicating that removing model identity may reduce bias and improve reasoning consistency.

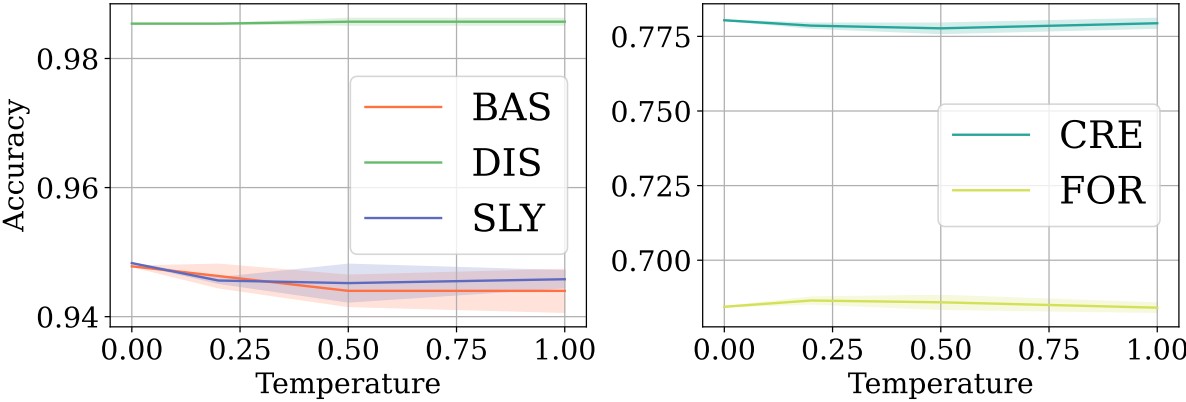

Figure 8: Effect of temperature setting on CoT²'s performance. We evaluate four values: $t = 0.1, 0.2, 0.5$, and 1.0. Results show that CoT² is generally robust to temperature changes, with performance remaining stable across different $t$ values. However, higher temperatures lead to increased variance, indicating less stable behavior from the LLM during inference.

## C   Detailed Results

Table 4: The detailed results shown in Figure 3.

| Dataset | CoT$^2$-Deepseek-v3 | MetaXGB | CoT$^2$-gpt-3.5 | Dataset | CoT$^2$-Deepseek-v3 | MetaXGB | CoT$^2$-gpt-3.5 |
|---|---|---|---|---|---|---|---|
| BAS | 94.55 | 95.52 | 94.63 | WPE | 92.03 | 88.93 | 91.14 |
| PC3 | 89.14 | 89.14 | 89.39 | ADH | 74.22 | 71.12 | 69.65 |
| MFF | 87.50 | 86.25 | 88.25 | SEG | 93.29 | 92.64 | 93.81 |
| DIS | 98.68 | 98.15 | 98.54 | GGI | 60.40 | 56.21 | 59.64 |
| WQW | 63.59 | 63.67 | 63.84 | RL | 78.81 | 77.67 | 77.87 |
| W5A | 85.80 | 83.30 | 86.12 | KDD | 81.15 | 78.35 | 80.10 |
| SYL | 94.93 | 94.05 | 94.56 | PBA | 97.44 | 96.53 | 97.50 |
| VUL | 98.95 | 98.95 | 98.95 | SAT | 92.40 | 90.75 | 92.40 |
| TBP | 97.20 | 96.41 | 97.27 | EMB | 62.67 | 59.86 | 62.65 |
| PEN | 99.43 | 99.18 | 99.45 | OSN | 90.30 | 89.94 | 90.18 |
| LSA | 100.0 | 100.0 | 100.0 |  |  |  |  |
| BWR | 98.74 | 98.63 | 98.70 | TTT | 81.47 | 78.54 | 81.52 |
| FOR | 70.69 | 66.53 | 68.65 | AHH | 68.55 | 66.34 | 65.65 |
| JC2 | 95.26 | 98.57 | 90.13 | CMC | 58.88 | 55.48 | 58.84 |
| CDD | 73.48 | 70.84 | 73.46 | YEA | 60.88 | 58.22 | 60.94 |

Table 5: RMSE on 15 regression datasets in TinyBench2. We report the RMSE of all the external models for each dataset. CoT$^2$ achieved the highest average ranking among all methods.

| Dataset | KNN | XGBoost | Catboost | LightGBM | MLP | ResNet | AutoInt | FT-T | Average | CoT$^2$ |
|---|---|---|---|---|---|---|---|---|---|---|
| ARC$_{\times 10^2}$ | 3.6422 | 3.3812 | 3.2327 | 3.4980 | 3.6477 | 3.5902 | 3.7367 | 4.0321 | 3.2382 | 3.2491 |
| HIE$_{\times 10^3}$ | 5.5246 | 4.6865 | 4.5222 | 4.6913 | 4.8525 | 4.7755 | 4.8049 | 4.5223 | 4.5460 | 4.6150 |
| CAC$_{\times 10^{-1}}$ | 1.3446 | 1.3502 | 1.2977 | 1.3308 | 1.3584 | 1.4602 | 1.3649 | 1.3791 | 1.3033 | 1.2989 |
| IGE$_{\times 10^{-2}}$ | 8.4527 | 4.2323 | 3.6572 | 4.2587 | 3.0002 | 2.4307 | 2.8165 | 3.0886 | 2.9917 | 2.7597 |
| KIN$_{\times 10^{-1}}$ | 1.2049 | 1.249 | 0.9029 | 1.2599 | 0.7488 | 7.3773 | 7.0919 | 0.6754 | 0.7835 | 0.7699 |
| BAN$_{\times 10^{-2}}$ | 4.9246 | 3.0842 | 2.8628 | 3.0073 | 2.8947 | 2.8571 | 2.8360 | 2.8245 | 2.8505 | 2.8142 |
| AIL$_{\times 10^{-4}}$ | 2.0400 | 1.5300 | 1.4700 | 1.5200 | 1.5500 | 1.5500 | 1.5500 | 1.5700 | 1.4800 | 1.4700 |
| LSW$_{\times 10^4}$ | 1.1759 | 0.5011 | 0.4449 | 0.4991 | 0.4841 | 0.5963 | 0.6354 | 0.4007 | 0.4249 | 0.3964 |
| COO$_{\times 10^0}$ | 1.4921 | 1.4795 | 1.4877 | 1.4833 | 1.5112 | 1.5921 | 1.5777 | 1.5899 | 1.4947 | 1.4951 |
| SUP$_{\times 10^1}$ | 1.0713 | 0.9959 | 0.9980 | 1.0103 | 1.0738 | 1.0365 | 1.0924 | 1.0593 | 0.9744 | 0.9754 |
| H1R$_{\times 10^4}$ | 3.7025 | 3.1061 | 3.0191 | 3.1017 | 3.1441 | 3.1448 | 3.1296 | 3.1265 | 2.9075 | 2.9143 |
| MV$_{\times 10^{-1}}$ | 15.1106 | 0.9397 | 0.8157 | 0.9257 | 0.2590 | 1.2554 | 0.4128 | 0.2684 | 1.9614 | 0.4875 |
| FRI$_{\times 10^0}$ | 1.8540 | 1.0838 | 1.0105 | 1.0627 | 1.0840 | 1.0230 | 1.0201 | 1.0100 | 1.0330 | 1.0122 |
| QFT$_{\times 10^{-1}}$ | 9.7412 | 9.2242 | 8.7159 | 8.9754 | 9.1436 | 9.4800 | 9.1662 | 9.1038 | 8.6904 | 8.9618 |
| C2R$_{\times 10^2}$ | 5.7960 | 4.6258 | 4.7438 | 4.7587 | 5.3895 | 5.1601 | 5.3207 | 5.3627 | 4.9601 | 4.9860 |
| average rank | 8.80 | 5.73 | 3.27 | 5.53 | 6.80 | 6.67 | 6.27 | 5.27 | 3.67 | 3.00 |

## D    Examples

```
Prompt

You are a machine learning expert that performs binclass task prediction.
I will give you a target instance, and I need you to make the final prediction decision based on a comprehensive analysis
of its neighbors and the predictions from some external trained models.

First, the information about the dataset:
Labels are integers ranging from 0 to 1. Class distribution: {0: 0.5, 1: 0.5}.

Second, the information about the external models:
The training accuracy of the models is as follows:
knn: 0.6934; xgboost: 0.8020; catboost: 0.7850; lightgbm: 0.8168;
mlp: 0.7585; resnet: 0.7573; autoint: 0.7585; ftt: 0.7523;
The validation accuracy of the models is as follows:
knn: 0.6654; xgboost: 0.7809; catboost: 0.7806; lightgbm: 0.7764;
mlp: 0.7574; resnet: 0.7589; autoint: 0.7563; ftt: 0.7604;

Third, the information about the neighbors, ordered from nearest to farthest:
No. 1, True label is 0;
[knn pred: 0]  [xgboost pred: 0]  [catboost pred: 0]  [lightgbm pred: 0]
[mlp pred: 0]  [resnet pred: 0]  [autoint pred: 0]  [ftt pred: 0]
No. 2, True label is 0;
[knn pred: 0]  [xgboost pred: 0]  [catboost pred: 0]  [lightgbm pred: 0]
[mlp pred: 0]  [resnet pred: 0]  [autoint pred: 0]  [ftt pred: 0]
No. 3, True label is 1;
[knn pred: 0]  [xgboost pred: 0]  [catboost pred: 1]  [lightgbm pred: 1]
[mlp pred: 0]  [resnet pred: 0]  [autoint pred: 0]  [ftt pred: 0]
No. 4, True label is 0;
[knn pred: 0]  [xgboost pred: 1]  [catboost pred: 0]  [lightgbm pred: 1]
[mlp pred: 0]  [resnet pred: 0]  [autoint pred: 0]  [ftt pred: 0]
No. 5, True label is 1;
[knn pred: 0]  [xgboost pred: 0]  [catboost pred: 0]  [lightgbm pred: 1]
[mlp pred: 0]  [resnet pred: 0]  [autoint pred: 0]  [ftt pred: 0]
No. 6, True label is 0;
[knn pred: 0]  [xgboost pred: 1]  [catboost pred: 1]  [lightgbm pred: 1]
[mlp pred: 0]  [resnet pred: 0]  [autoint pred: 0]  [ftt pred: 0]
No. 7, True label is 0;
[knn pred: 0]  [xgboost pred: 0]  [catboost pred: 0]  [lightgbm pred: 0]
[mlp pred: 0]  [resnet pred: 0]  [autoint pred: 0]  [ftt pred: 0]
No. 8, True label is 1;
[knn pred: 1]  [xgboost pred: 1]  [catboost pred: 1]  [lightgbm pred: 1]
[mlp pred: 1]  [resnet pred: 1]  [autoint pred: 1]  [ftt pred: 1]
No. 9, True label is 0;
[knn pred: 0]  [xgboost pred: 0]  [catboost pred: 0]  [lightgbm pred: 0]
[mlp pred: 0]  [resnet pred: 0]  [autoint pred: 0]  [ftt pred: 0]
No. 10, True label is 0;
[knn pred: 0]  [xgboost pred: 0]  [catboost pred: 0]  [lightgbm pred: 0]
[mlp pred: 0]  [resnet pred: 0]  [autoint pred: 0]  [ftt pred: 0]

Fourth, the information about the target instance:
[knn pred: 0]  [xgboost pred: 1]  [catboost pred: 1]  [lightgbm pred: 1]
[mlp pred: 0]  [resnet pred: 0]  [autoint pred: 0]  [ftt pred: 0]

Let's think step by step:

1. Based on the training accuracies and validation accuracies of each model, You infer the overall performance of the
external models on the dataset. Then you select well-performing models from external models. We aim for you to identify
overfitting and underfitting models based on their training and validation accuracies, and to find the overall
well-performing models dataset.

2. Based on the true labels of the neighbors, the neighbors' predicted labels from well-performing models, and the label
frequencies, you identify non-outliers among the neighbors. If the majority of well-performing models predict incorrectly
for a particular neighbor, it suggests that this neighbor might be an outlier, negatively affecting the predictions. We
want you to be able to identify such outliers. Label frequencies provide additional information about the degree of data
imbalance, which aids in reasoning.

3. Based on the true labels of the non-outliers, the non-outliers' predicted labels from all models, and the label
frequencies, you select the most suitable models for the neighborhood space of the target instance. Models that perform
well overall on the dataset may not be the most efficient at predicting the target instance. It is essential to identify
the best-suited models for the target instance within the neighbor space after filtering out outliers.

4. Based on the true labels of the non-outlier neighbors, the label frequencies, and the target instance's predicted labels
from the most suitable models and well-performing models, list the labels of the non-outlier neighbors and the predictions
of the most suitable models on the target instance. This will help you assess the effectiveness of these models in the
target instance's neighborhood. After removing outliers and unsuitable external models, you can use a KNN-based approach
and model ensembling within the clean local context to achieve the most confident final predictions. Well-performing models,
being the strongest models on the current dataset leaderboard, provide auxiliary information for the final prediction.

I will use Python code to extract your prediction. Please ensure your response allows the following code to successfully
obtain your predicted label:
import re
label = re.search(r'I predict the label of the target instance as (\d+)', your_response_text).group(1)

To match the regex, your response must strictly contain this sentence after your reasoning steps:
"I predict the label of the target instance as [Your Answer]."
```

Figure 9: An example of the prompt in the classification dataset CRE. We also provided examples in Figure 12 and Figure 11 where gpt-3.5-turbo responds without and with CoT², respectively. It can be observed that CoT² breaks down a complex problem into multiple steps, resulting in more structured answers, thus enhancing the interpretability and accuracy. The responses of Deepseek-v3 and GPT-4o are in Figure 14 and Figure 13. We further provide step-by-step responses from the latest ChatGPT to illustrate the reasoning process in more detail, as shown in Figure 15, 16, 17, and 18.

**Prompt**

You are a machine learning expert that performs regression task prediction.
I will give you a target instance, and I need you to make the final prediction decision based on a comprehensive analysis of its neighbors and the predictions from some external trained models.

First, the information about the dataset:
Labels range from -1.0058 to 4.4005.

Second, the information about the external models:
The training RMSE of the models is as follows:
knn: 0.1216; xgboost: 0.1345; catboost: 0.1420; lightgbm: 0.1575;
mlp: 0.2364; resnet: 0.2215; autoint: 0.2275; ftt: 0.2161;
The validation RMSE of the models is as follows:
knn: 0.2876; xgboost: 0.2635; catboost: 0.2588; lightgbm: 0.2628;
mlp: 0.2931; resnet: 0.2823; autoint: 0.2952; ftt: 0.2838;

Third, the information about the neighbors, ordered from nearest to farthest:
No. 1, True label is -0.9582;
[knn pred: -0.9582]  [xgboost pred: -0.9504]  [catboost pred: -0.9638]  [lightgbm pred: -0.9420]
[mlp pred: -0.9549]  [resnet pred: -1.0089]  [autoint pred: -0.9115]  [ftt pred: -0.9284]
No. 2, True label is -0.9804;
[knn pred: -0.9683]  [xgboost pred: -0.9504]  [catboost pred: -0.9652]  [lightgbm pred: -0.9420]
[mlp pred: -0.9551]  [resnet pred: -1.0077]  [autoint pred: -0.9088]  [ftt pred: -0.9237]
No. 3, True label is -0.9561;
[knn pred: -0.9683]  [xgboost pred: -0.9504]  [catboost pred: -0.9652]  [lightgbm pred: -0.9420]
[mlp pred: -0.9551]  [resnet pred: -1.0077]  [autoint pred: -0.9088]  [ftt pred: -0.9237]
No. 4, True label is -0.9813;
[knn pred: -0.9813]  [xgboost pred: -0.9526]  [catboost pred: -0.9711]  [lightgbm pred: -0.9413]
[mlp pred: -0.9393]  [resnet pred: -1.0271]  [autoint pred: -0.8666]  [ftt pred: -0.8721]
No. 5, True label is -0.9620;
[knn pred: -0.9620]  [xgboost pred: -0.9409]  [catboost pred: -0.9481]  [lightgbm pred: -0.9510]
[mlp pred: -0.9424]  [resnet pred: -0.9613]  [autoint pred: -0.8746]  [ftt pred: -0.8476]
No. 6, True label is -0.8159;
[knn pred: -0.8159]  [xgboost pred: -0.7953]  [catboost pred: -0.8107]  [lightgbm pred: -0.8180]
[mlp pred: -0.8514]  [resnet pred: -0.8512]  [autoint pred: -0.8824]  [ftt pred: -0.8675]
No. 7, True label is -0.7954;
[knn pred: -0.7954]  [xgboost pred: -0.8337]  [catboost pred: -0.8085]  [lightgbm pred: -0.8160]
[mlp pred: -0.8598]  [resnet pred: -0.7583]  [autoint pred: -0.9357]  [ftt pred: -0.9902]
No. 8, True label is -0.9123;
[knn pred: -0.9123]  [xgboost pred: -0.9207]  [catboost pred: -0.9147]  [lightgbm pred: -0.9185]
[mlp pred: -0.9698]  [resnet pred: -0.8695]  [autoint pred: -0.9723]  [ftt pred: -0.9036]
No. 9, True label is -0.8217;
[knn pred: -0.8217]  [xgboost pred: -0.8567]  [catboost pred: -0.8416]  [lightgbm pred: -0.8620]
[mlp pred: -0.8576]  [resnet pred: -0.8030]  [autoint pred: -0.8840]  [ftt pred: -0.8008]
No. 10, True label is -0.6259;
[knn pred: -0.6259]  [xgboost pred: -0.6181]  [catboost pred: -0.6195]  [lightgbm pred: -0.6031]
[mlp pred: -0.6648]  [resnet pred: -0.6208]  [autoint pred: -0.6959]  [ftt pred: -0.7828]

Fourth, the information about the target instance:
[knn pred: -0.9605]  [xgboost pred: -0.9504]  [catboost pred: -0.9592]  [lightgbm pred: -0.9420]
[mlp pred: -0.9549]  [resnet pred: -1.0090]  [autoint pred: -0.9119]  [ftt pred: -0.9293]

Let's think step by step:

1. Based on the training accuracies and validation accuracies of each model, You infer the overall performance of the external models on the dataset. Then you select well-performing models from external models. We aim for you to identify overfitting and underfitting models based on their training and validation accuracies, and to find the overall well-performing models dataset.

2. Based on the true labels of the neighbors, the neighbors' predicted labels from well-performing models, and the label frequencies, you identify non-outliers among the neighbors. If the majority of well-performing models predict incorrectly for a particular neighbor, it suggests that this neighbor might be an outlier, negatively affecting the predictions. We want you to be able to identify such outliers. Label frequencies provide additional information about the degree of data imbalance, which aids in reasoning.

3. Based on the true labels of the non-outliers, the non-outliers' predicted labels from all models, and the label frequencies, you select the most suitable models for the neighborhood space of the target instance. Models that perform well overall on the dataset may not be the most efficient at predicting the target instance. It is essential to identify the best-suited models for the target instance within the neighbor space after filtering out outliers.

4. Based on the true labels of the non-outlier neighbors, the label frequencies, and the target instance's predicted labels from the most suitable models and well-performing models, list the labels of the non-outlier neighbors and the predictions of the most suitable models on the target instance. This will help you assess the effectiveness of these models in the target instance's neighborhood. After removing outliers and unsuitable external models, you can use a KNN-based approach and model ensembling within the clean local context to achieve the most confident final predictions. Well-performing models, being the strongest models on the current dataset leaderboard, provide auxiliary information for the final prediction.

I will use Python code to extract your prediction. Please ensure your response allows the following code to successfully obtain your predicted label:
import re
label = re.search(r'I predict the label of the target instance as (\d+)', your_response_text).group(1)

To match the regex, your response must strictly contain this sentence after your reasoning steps:
"I predict the label of the target instance as [Your Answer]."

Figure 10: An example of the prompt in the regression dataset KIN.

**Response**

1. Based on the training and validation accuracies of the external models, we can infer the overall performance on the dataset. The models with high training and validation accuracies such as lightgbm (0.8168 training, 0.7764 validation) and xgboost (0.8020 training, 0.7809 validation) seem to perform well without overfitting or underfitting.

2. Identifying non-outliers among the neighbors:
   - No. 1, 2, 4, 6, 7, 9, 10 are non-outliers as they have a majority of correct predictions from well-performing models.
   - No. 3, 5, 8 are outliers as most of the well-performing models predict incorrectly for them.

3. Selecting the most suitable models for the target instance:
   Based on the non-outlier neighbors, the models that seem suitable for the target instance are knn, resnet, and ftt.

4. Final analysis:
   - Non-outlier neighbors' labels: 0, 0, 0, 0, 0, 0, 0, 0, 0, 0
   - Predictions from the most suitable models on the target instance:
     [knn pred: 0]   [resnet pred: 0]   [ftt pred: 0]

Considering the majority label from the non-outlier neighbors and the predictions from the most suitable models,
the final prediction for the target instance is label 0.

I predict the label of the target instance as 0.

Figure 11: The response from gpt-3.5-turbo with CoT$^2$ for Figure 9, where the true label corresponding to the question is 0, and the model predicted 0, indicating a right prediction.

**Response**

Based on the information provided, we can analyze the predictions from the nearest neighbors and the external models to make a final prediction for the target instance.

Looking at the nearest neighbors, we see that the majority of them have predicted the target instance as label 0, except for one neighbor which predicted it as label 1. This indicates some uncertainty in the neighborhood around the target instance.

Considering the predictions from the external models, we see that xgboost, catboost, and lightgbm have predicted the target instance as label 1, while knn, mlp, resnet, autoint, and ftt have predicted it as label 0. The models with higher validation accuracy (xgboost, catboost, lightgbm) are leaning towards predicting the target instance as label 1.

Given the mix of predictions from the neighbors and the external models, and considering the higher validation accuracy of xgboost, catboost, and lightgbm, we can make a final prediction that the target instance is likely to be label 1.

Therefore, based on the analysis, I predict the label of the target instance as 1.

Figure 12: The response from gpt-3.5-turbo without CoT$^2$ for Figure 9, where the true label corresponding to the question is 0, and the model predicted 1, indicating a right prediction.

**Response**

To predict the label of the target instance, let's go through the steps outlined:

1. **Evaluate External Models:**
   - **Training vs. Validation Accuracy:**
     - **KNN:** Training: 0.6934, Validation: 0.6654 (Slight overfitting)
     - **XGBoost:** Training: 0.8020, Validation: 0.7809 (Good performance, slight overfitting)
     - **CatBoost:** Training: 0.7850, Validation: 0.7806 (Good performance, minimal overfitting)
     - **LightGBM:** Training: 0.8168, Validation: 0.7764 (Overfitting)
     - **MLP:** Training: 0.7585, Validation: 0.7574 (Good balance)
     - **ResNet:** Training: 0.7573, Validation: 0.7589 (Good balance)
     - **AutoInt:** Training: 0.7585, Validation: 0.7563 (Good balance)
     - **FTT:** Training: 0.7523, Validation: 0.7604 (Good balance)
   - **Well-performing Models:** XGBoost, CatBoost, MLP, ResNet, AutoInt, FTT
     (due to good validation performance and minimal overfitting)

2. **Identify Non-Outliers Among Neighbors:**
   - **Neighbor Analysis:**
     - Neighbors 1, 2, 4, 6, 7, 9, 10 have true label 0.
     - Neighbors 3, 5, 8 have true label 1.
   - **Outlier Detection:**
     - Neighbor 8 is an outlier as all models predict 1, but most neighbors are labeled 0.
     - Neighbors 3 and 5 are also potential outliers as they have mixed predictions and true label 1,
       while most neighbors are 0.

3. **Select Suitable Models for Neighborhood:**
   - **Non-Outlier Neighbors:** 1, 2, 4, 6, 7, 9, 10 (all true label 0)
   - **Model Performance on Non-Outliers:**
     - XGBoost, CatBoost, LightGBM show some incorrect predictions on non-outliers.
     - MLP, ResNet, AutoInt, FTT consistently predict 0 for non-outliers.
   - **Most Suitable Models:** MLP, ResNet, AutoInt, FTT

4. **Final Prediction Using KNN and Model Ensembling:**
   - **Non-Outlier Neighbors' True Labels:** All are 0.
   - **Target Instance Predictions from Suitable Models:**
     - MLP: 0, ResNet: 0, AutoInt: 0, FTT: 0
   - **Well-Performing Models' Predictions on Target:**
     - XGBoost: 1, CatBoost: 1, MLP: 0, ResNet: 0, AutoInt: 0, FTT: 0
   - **KNN-Based Approach:** Majority of non-outlier neighbors are 0.
   - **Model Ensembling:** Majority of suitable models predict 0.

Given the analysis, the majority of the non-outlier neighbors have a true label of 0, and the most suitable models for
the neighborhood also predict 0 for the target instance. Therefore, the final prediction for the target instance is:

I predict the label of the target instance as 0.

Figure 13: The response from gpt-4o for Figure 9, where the true label corresponding to the question is 0, and the model predicted 0, indicating a right prediction.

**Response**

### Step 1: Infer the overall performance of the external models and select well-performing models

The training and validation accuracies of the models are as follows:

- **knn**: Training: 0.6934, Validation: 0.6654
- **xgboost**: Training: 0.8020, Validation: 0.7809
- **catboost**: Training: 0.7850, Validation: 0.7806
- **lightgbm**: Training: 0.8168, Validation: 0.7764
- **mlp**: Training: 0.7585, Validation: 0.7574
- **resnet**: Training: 0.7573, Validation: 0.7589
- **autoint**: Training: 0.7585, Validation: 0.7563
- **ftt**: Training: 0.7523, Validation: 0.7604

**Analysis:**
- **Overfitting Models**: Models with a significant gap between training and validation accuracy
(e.g., lightgbm, xgboost) may be overfitting.
- **Underfitting Models**: Models with low training and validation accuracy (e.g., knn) may be underfitting.
- **Well-performing Models**: Models with high and close training and validation accuracy
(e.g., catboost, mlp, resnet, autoint, ftt) are considered well-performing.

**Selected Well-performing Models**: catboost, mlp, resnet, autoint, ftt

### Step 2: Identify non-outliers among the neighbors

**Neighbors' True Labels and Predictions:**

| Neighbor | True Label | catboost | mlp | resnet | autoint | ftt |
|----------|------------|----------|-----|--------|---------|-----|
| 1        | 0          | 0        | 0   | 0      | 0       | 0   |
| 2        | 0          | 0        | 0   | 0      | 0       | 0   |
| 3        | 1          | 1        | 0   | 0      | 0       | 0   |
| 4        | 0          | 0        | 0   | 0      | 0       | 0   |
| 5        | 1          | 0        | 0   | 0      | 0       | 0   |
| 6        | 0          | 1        | 0   | 0      | 0       | 0   |
| 7        | 0          | 0        | 0   | 0      | 0       | 0   |
| 8        | 1          | 1        | 1   | 1      | 1       | 1   |
| 9        | 0          | 0        | 0   | 0      | 0       | 0   |
| 10       | 0          | 0        | 0   | 0      | 0       | 0   |

**Outlier Identification:**
- **Neighbor 3**: Majority of well-performing models predict 0, but the true label is 1.
- **Neighbor 5**: Majority of well-performing models predict 0, but the true label is 1.
- **Neighbor 6**: Majority of well-performing models predict 0, but the true label is 0.
    However, catboost predicts 1, which is incorrect.
This suggests that Neighbor 3, 5, 6 might be outliers.

**Non-outliers**: Neighbors 1, 2, 4, 7, 8, 9, 10

### Step 3: Select the most suitable models for the neighborhood space of the target instance

**Non-outliers' True Labels and Predictions:**

| Neighbor | True Label | catboost | mlp | resnet | autoint | ftt |
|----------|------------|----------|-----|--------|---------|-----|
| 1        | 0          | 0        | 0   | 0      | 0       | 0   |
| 2        | 0          | 0        | 0   | 0      | 0       | 0   |
| 4        | 0          | 0        | 0   | 0      | 0       | 0   |
| 7        | 0          | 0        | 0   | 0      | 0       | 0   |
| 8        | 1          | 1        | 1   | 1      | 1       | 1   |
| 9        | 0          | 0        | 0   | 0      | 0       | 0   |
| 10       | 0          | 0        | 0   | 0      | 0       | 0   |

**Analysis:**
- **catboost**: Correctly predicts all non-outliers.
- **mlp**: Correctly predicts all non-outliers.
- **resnet**: Correctly predicts all non-outliers.
- **autoint**: Correctly predicts all non-outliers.
- **ftt**: Correctly predicts all non-outliers.

**Most Suitable Models**: catboost, mlp, resnet, autoint, ftt

### Step 4: Make the final prediction

**Target Instance Predictions from Most Suitable Models:**

| Model    | Prediction |
|----------|------------|
| catboost | 1          |
| mlp      | 0          |
| resnet   | 0          |
| autoint  | 0          |
| ftt      | 0          |

**Analysis:**
- **Majority Vote**: 0 (mlp, resnet, autoint, ftt) vs 1 (catboost)
- **KNN-based Approach**: Considering the non-outlier neighbors, the majority label is 0.

**Final Prediction**: 0

I predict the label of the target instance as 0.

Figure 14: The response from DeepSeek-v3 for Figure 9, where the true label corresponding to the question is 0, and the model predicted 0, indicating a right prediction.

## Step 1: Evaluate External Models' Performance

We analyze both training and validation accuracies to assess overfitting and generalization:

| Model | Train Acc | Val Acc | Gap (Train - Val) |
|---|---|---|---|
| knn | 0.6934 | 0.6654 | 0.0280 |
| xgboost | 0.8020 | 0.7809 | 0.0211 |
| catboost | 0.7850 | 0.7806 | 0.0044 |
| lightgbm | 0.8168 | 0.7764 | 0.0404 |
| mlp | 0.7585 | 0.7574 | 0.0011 |
| resnet | 0.7573 | 0.7589 | -0.0016 |
| autoint | 0.7585 | 0.7563 | 0.0022 |
| ftt | 0.7523 | 0.7604 | -0.0081 |

Models with high validation accuracy and low train-val gap are considered well-performing. Let's select models with validation accuracy ≥ 0.76 and low overfitting:

- ✅ **xgboost** (0.7809)
- ✅ **catboost** (0.7806)
- ✅ **lightgbm** (0.7764)
- ✅ **ftt** (0.7604)

These are our **well-performing models**.

Figure 15: Step 1 of the response generated by the latest ChatGPT model for the example in Figure 9. This step demonstrates how the model begins to decompose the problem and reason through the input systematically.

**Step 2: Identify Non-Outlier Neighbors**

We now look at the predictions of well-performing models on each neighbor and compare them to the true label.

Neighbors:

1. Label 0 — All well-performing models predicted 0 → ✅ correct

2. Label 0 — All well-performing models predicted 0 → ✅ correct

3. Label 1 — xgboost: 0 ❌, catboost: 1 ✅, lightgbm: 1 ✅, ftt: 0 ❌ → 2 correct, 2 incorrect → borderline

4. Label 0 — xgboost: 1 ❌, catboost: 0 ✅, lightgbm: 1 ❌, ftt: 0 ✅ → 2 correct, 2 incorrect → borderline

5. Label 1 — xgboost: 0 ❌, catboost: 0 ❌, lightgbm: 1 ✅, ftt: 0 ❌ → 1 correct, 3 incorrect → likely ❌ outlier

6. Label 0 — xgboost: 1 ❌, catboost: 1 ❌, lightgbm: 1 ❌, ftt: 0 ✅ → 1 correct, 3 incorrect → ❌ outlier

7. Label 0 — All correct → ✅

8. Label 1 — All correct → ✅

9. Label 0 — All correct → ✅

10. Label 0 — All correct → ✅

**Outliers:** Neighbors 5 and 6
**Borderline (keep for now):** Neighbors 3 and 4
**Non-outliers:** Neighbors 1, 2, 3, 4, 7, 8, 9, 10

Figure 16: Step 2 of the response generated by the latest ChatGPT model for the example in Figure 9.

## Step 3: Select Most Suitable Models in the Neighborhood

Now we evaluate how each model performs on the non-outlier neighbors (Neighbors 1,2,3,4,7,8,9,10).

Let's count correct predictions for each model on these 8 neighbors:

| Model | Correct Predictions |
| --- | --- |
| knn | 7 (missed neighbor 3) |
| xgboost | 5 (missed 3,4,9) |
| catboost | 7 (missed 5, not in non-outliers) |
| lightgbm | 6 (missed 4,9) |
| mlp | 7 (missed 3) |
| resnet | 7 (missed 3) |
| autoint | 7 (missed 3) |
| ftt | 6 (missed 3,4) |

So, models that perform best in this neighborhood:

- ✅ knn (7/8)
- ✅ catboost (7/8)
- ✅ mlp (7/8)
- ✅ resnet (7/8)
- ✅ autoint (7/8)

These are the most suitable models for the neighborhood.

Figure 17: Step 3 of the response generated by the latest ChatGPT model for the example in Figure 9.

**Step 4: Final Prediction Using Local and Global Evidence**

Let's now look at:

- True labels of non-outlier neighbors:
  Neighbors 1,2,4,7,9,10 → label 0
  Neighbors 3,8 → label 1
  → Class distribution in local space: 6 of 8 are class 0 → strong local bias toward class 0

- Predictions of the most suitable models on the target instance:

  - knn: 0, catboost: 1, mlp: 0, resnet: 0, autoint: 0
    → 4 votes for label 0, 1 vote for label 1

- Predictions of the well-performing models:

  - xgboost: 1, catboost: 1, lightgbm: 1, ftt: 0
    → 3 votes for label 1, 1 for label 0

**Interpretation:**

- Most suitable models in the local neighborhood strongly favor label 0 (4 to 1).
- Well-performing models slightly favor label 1 (3 to 1), but these models are global and may not reflect local structure.
- The local neighborhood has a strong majority of class 0 (6/8).
- The most suitable models have a high accuracy in the neighborhood and favor class 0.

✅ **Final Decision:**

Given the strong local evidence from both neighbor labels and the most suitable models, we prioritize the local context. The slight disagreement from global models is not enough to override the strong local consensus.

**I predict the label of the target instance as 0.**

Figure 18: Step 4 of the response generated by the latest ChatGPT model for the example in Figure 9.

