# OpenReview forum: "Make Still Further Progress: Chain of Thoughts for Tabular Data Leaderboard"
_TMLR — Under review for TMLR_

### Review · Reviewer_YSmP · 2026-05-19

**Summary Of Contributions:**

## Summary:

The paper introduces an in-context ensemble framework for tabular data prediction called Chain of Tabular Thoughts. Rather than converting raw tabular features into text prompts or automating AutoML pipelines, the authors frame a LLM as an instance-level ensembling expert. For each test instance, a local tabular context is constructed by retrieving its $K$-nearest neighbors (using mutual-information re-weighted distances) and compiling the true neighbor labels along with out-of-sample predictions from a pool of base models (such as XGBoost, LightGBM, CatBoost, and several tabular neural networks). The LLM is then guided via a four-step reasoning prompt ($CoT^2$), consisting of well-performing model selection, outlier identification, suitable local model selection, and final prediction,to infer the target label. To mitigate inference overhead, the framework uses a hard-sample identification heuristic to route instances with high base-model consensus away from the LLM.

-----

## Key Strengths:
- The method does not expose raw feature text or semantic metadata to the LLM, making it potentially applicable to fully anonymized or unannotated tabular schemas.
- Instance-Aware Ensembling: It shifts the use of LLMs in tabular data from basic serialization/prediction to a meta-learning/ensembling agent that operates dynamically over local performance signals.
- Cost-Aware Routing: The inclusion of a cascading consensus routing mechanism (bypassing the LLM for "easy" samples) shows an awareness of the practical costs associated with LLM calls.


## Key Weaknesses:
- Literature Over-simplification and Outdated Baselines: The paper bases its core motivation on the premise that GBDTs consistently dominate deep learning models on tabular benchmarks. This ignores major modern shifts in the tabular ecosystem (e.g., TabArena benchmarks), where recent deep architectures significantly outperform standard GBDTs.

- Omission of State-of-the-Art Pre-trained/Tabular Foundation Models: While the paper positions itself within retrieval-augmented and in-context tabular methods, it completely omits comparison with modern Prior-Data Networks (PFNs) and In-Context Learning architectures like TabPFN (v3), TabICLv2, ...

- Absence of Standard Tabular Ensemble Baselines: For a work focusing entirely on model ensembling, the omission of AutoGluon is questionable.

Corrupted / Compromised Citation Pipeline: The bibliography and text display systematic citation errors, suggesting an automated or unverified assembly of references.

**Audience:**

Yes

**Audience Explanation:**

Conceptually, treating Large Language Models as black-box meta-ensemblers over local neighbor contexts is an intriguing direction. Researchers working at the intersection of tabular deep learning, foundation models, and privacy-preserving machine learning will find the problem formulation and the hard-sample routing strategy of interest. However, the audience will only find the findings valuable if they are grounded against the actual state-of-the-art in tabular data mining.

**Claims And Evidence:**

No

**Claims Explanation:**

The empirical evidence presented in the paper fails to justify the claim of achieving "expert-level competition performance" or outperforming standard ensemble techniques due to severe gaps in baseline selection and benchmarking context:

1. The authors rely heavily on older literature (e.g., Grinsztajn et al., 2022) to assert that GBDTs are overwhelmingly superior to deep tabular architectures. However, on modern tabular benchmarks such as TabArena, models like TabM and RealMLP routinely outperform standard GBDTs. By relying on an unrepresentative baseline landscape, the relative gains of the proposed meta-ensemble are called into question.
2. Complete Omission of Modern PFN / ICL Baselines: In Section 2.3, the authors discuss Retrieval-Augmented Generation and briefly touch upon the foundational TabPFN (Hollmann et al., 2025). Yet, they do not include or compare against modern iterations like TabPFN-V3 or TabICLv2. Given that these architectures are specifically designed for zero-shot in-context classification and regression on tabular data, they are direct competitors to an LLM doing meta-predictions on neighbor pools.
3. Missing Critical Ensemble Baselines (AutoGluon): The core contribution is an instance-wise ensembling agent. However, the paper compares its approach against simple voting, averaging, or a rudimentary custom XGBoost meta-learner (MetaXGB). It completely omits AutoGluon. Without outperforming AutoGluon, the claim that $CoT^2$ provides superior ensemble execution remains unsupported.

**Requested Changes:**

1. Incorporate Missing Baselines on TinyBench2: Please include AutoGluon (1.5, extreme)  as a core baseline in your ensemble comparisons.

2. benchmark against modern foundation models that reflect current performance frontiers, specifically TabICLv2 and TabPFNv2+

3. Update the Literature Review and Evaluation Framing: Rewrite sections of the Introduction and Related Work to move away from the outdated narrative that "GBDTs always win"

4. Fix the broken inline citations

5. Report actual wall-clock time of the method compared to other methods, for training and inference.

---

### Review · Reviewer_AUrD · 2026-05-24

**Summary Of Contributions:**

This paper proposes Chain of Tabular Thought ($CoT^2$) , a novel strategy for improving prediction performance on tabular data using large language models. The proposed approach can be viewed as a reasoning-based ensemble framework that combines predictions from existing tabular methods while additionally leveraging the reasoning capability of LLMs.

More specifically, the method introduces a four-stage reasoning pipeline designed to elicit “thinking” behaviors from LLMs through carefully constructed prompts. By integrating predictions from multiple methods and allowing the LLM to reason about potential inconsistencies and difficult samples, the approach aims to improve robustness and prediction accuracy on tabular tasks. Experimental results demonstrate performance improvements across several benchmarks, suggesting that the paper provides an interesting direction for utilizing LLM reasoning abilities in tabular learning problems.

Overall, I believe the paper presents a meaningful and practically relevant idea for the tabular learning community, especially because it explores how LLM reasoning capabilities can be effectively adapted beyond standard text-centric tasks.

**Audience:**

Yes

**Audience Explanation:**

Yes. I believe researchers working on tabular learning, LLM reasoning, and LLM applications beyond pure NLP tasks would find this work interesting. The paper explores an emerging and practically important question: how to effectively utilize the reasoning capability of LLMs for structured tabular prediction problems.

**Claims And Evidence:**

Yes

**Claims Explanation:**

The paper provides empirical evidence demonstrating that the proposed $CoT^2$ framework can improve prediction performance on several tabular benchmarks, and the experimental section is generally clear and reasonably organized. The results support the main claim that aggregation using LLMs with reasoning can be beneficial for tabular data tasks.

However, I do not find the evidence fully convincing yet for several reasons. First, the overall method is highly heuristic, and the paper currently lacks sufficient justification for why each reasoning stage is necessary or theoretically meaningful. While the intuition is understandable, the design choices appear somewhat ad hoc, making it difficult to assess whether the observed gains stem from the proposed reasoning framework itself or from increased inference complexity and ensembling effects.

Second, for strong LLMs such as GPT-4o and DeepSeek-V3, the performance gap between using and not using CoT^2 appears relatively small. This weakens the argument that the proposed reasoning framework consistently provides substantial benefits beyond standard prompting or direct prediction.

Finally, the paper would benefit from a more detailed analysis of success and failure cases. In particular, it would be valuable to discuss:

* in what types of tabular tasks the proposed reasoning framework is most effective,
* when nearest-neighbor or outlier verification helps,
* and in which situations the method provides limited gains or even fails.

Such analyses would significantly improve the clarity and credibility of the paper’s claims.

**Requested Changes:**

1. Provide stronger justification for each stage of the CoT^2 pipeline. The current method appears highly heuristic, and the paper would benefit from clearer motivation or analysis explaining why each reasoning step is necessary and how it contributes to performance improvements.

2. Include more detailed ablation studies analyzing the contribution of each component in the four-stage reasoning framework.

3. Discuss why the performance improvements for strong LLMs such as GPT-4o and DeepSeek-V3 are relatively limited in some settings. This point should be addressed more explicitly in the paper.

4. Add qualitative analyses and case studies illustrating:
    * successful examples where CoT^2 substantially improves predictions,
    * failure cases where the reasoning process does not help,
    * and characteristics of datasets or samples where the method is particularly effective or ineffective.

---

### Comment · Action_Editor_ibAV · 2026-06-26
**discussion**

Dear all,

First of all, let me thank the reviewers again for their service.  We are now in the discussion phase, and from a quick scan of the reviews, it looks like we have a reasonable plan for revision that would hopefully bring the paper above the bar of TMLR.  So let's try to address all outstanding issues and converge to a final decision in ~2 weeks.

Best,
Your AE